# Single-cell reconstruction reveals input patterns and pathways into corticotropin-releasing factor neurons in the central amygdala in mice

Chuan Huang [1,2✉], Yu Wang[1,2], Peng Chen[1,2], Qing-Hong Shan[1,2], Hao Wang[3,4], Lu-Feng Ding[1,2], Guo-Qiang Bi [1,2] & Jiang-Ning Zhou [1,2✉]

Corticotropin-releasing factor (CRF) neurons are one of the most densely distributed cell types in the central amygdala (CeA), and are involved in a wide range of behaviors including anxiety and learning. However, the fundamental input circuits and patterns of CeA-CRF neurons are still unclear. Here, we generate a monosynaptic-input map onto CeA-CRF neurons at single-cell resolution via a retrograde rabies-virus system. We find all inputs are located in 44 nested subregions that directly innervate CeA-CRF neurons; most of them are top-down convergent inputs expressing $Ca^{2+}$/calmodulin-dependent protein kinase II, and are centralized in cortex, especially in the layer 4 of the somatosensory cortex, which may directly relay information from the thalamus. While the bottom-up divergent inputs have the highest proportion of glutamate decarboxylase expression. Finally, *en passant* structures of single input neuron are revealed by in-situ reconstruction in a modified 3D-reference atlas, represented by a Periaqueductal gray-Subparafascicular nucleus-Subthalamic nucleus-Globus pallidus-Caudoputamen-CeA pathway. Taken together, our findings provide morphological and connectivity properties of inputs onto CeA-CRF neurons, which may provide insights for future studies interrogating circuit mechanisms of CeA-CRF neurons in mediating various functions.

[1] Hefei National Laboratory for Physical Sciences at the Microscale, Chinese Academy of Sciences Key Laboratory of Brain Function and Diseases, Division of Life Sciences and Medicine, University of Science and Technology of China, Hefei, PR China. [2] Center for Excellence in Brain Science and Intelligence Technology, Chinese Academy of Sciences, Shanghai, China. [3] National Engineering Laboratory for Brain-inspired Intelligence Technology and Application, University of Science and Technology of China, Hefei, China. [4] Institute of Artificial Intelligence, Hefei Comprehensive National Science Center, Hefei, China. ✉email: chuanh@ustc.edu.cn; jnzhou@ustc.edu.cn

The amygdala has been demonstrated to play a critical role in a range of brain functions—including emotions, learning, memory, attention, and perception—especially in terms of processing environmental stimuli associated with fear and reward[1]. It has been posited that the complex anatomy of the amygdala and its diverse neuronal subtypes confer a wide variety of important functions[2]. Furthermore, increased attention has recently been devoted to elucidating specific amygdalar circuits and their corresponding functions[1]. However, different regions of the amygdala have unique connections with other brain structures[3], and different molecularly defined neurons in the amygdala undertake distinct functions in various behaviors. The refined but basic neural circuits and input patterns to the amygdala remain unknown, and their identification may provide guidance for a better understanding of the behaviors in which the amygdala is involved.

The basolateral amygdala (BLA) and central amygdala (CeA) are two major nuclei that play essential roles in various behaviors. Substantial information processing occurs between the BLA and CeA. As the main integrated input nucleus into the amygdala, the BLA receives inputs from upstream loci and transmits this information to the CeA. In contrast, the CeA acts as the output nucleus of the amygdala, such that it innervates several downstream brain regions that enable the body to adaptively respond to external stimuli[4]. However, in addition to receiving information from the BLA, the CeA also receives direct inputs from both the thalamus and cortex, and contributes to the expression of innate behaviors and associated physiological responses[5].

The CeA comprises a wide array of molecularly distinct cell types, which play different roles in amygdala-mediated behaviors[6]. In addition to neuronal subtypes expressing either somatostatin (SST) or protein kinase C-δ (PKC-δ) in inhibitory circuits encoding fear[7], peptide-expressing neurons in the CeA have recently been investigated[8]. Corticotropin-releasing factor (CRF) is a stress-related peptide that is expressed in a large subpopulation of CeA neurons[9]. CRF acts as an endocrine factor to regulate stress responses via the hypothalamic–pituitary–adrenal axis, and acts as a neuromodulator in the central nervous system to regulate food intake, energy metabolism, and emotional responses[10–12]. Interestingly, CeA-CRF neurons represent one of the most densely distributed populations of CRF neurons throughout the brain[13], which has attracted the attention of many research groups[14,15]. By employing CRF-Cre mice or rats, it has been discovered that they are involved in mediating stress[16], pain[17], alcohol addiction[18], and fear[19,20]. In addition, by injecting the rabies virus, a fraction of putative excitatory input brain regions were retrogradely traced in previous research, which is consistent with the current data set[21]. However, so far there is no study that systematically elaborates the input of CeA-CRF neurons at the whole-brain scale.

To elucidate the functions of CeA-CRF neurons, it is necessary to comprehensively dissect their connectivity. The amygdala sends projections from the CeA to the stria terminalis, basal forebrain, various hypothalamic nuclei, midline thalamic nuclei, and the brainstem[22]. Major efferent pathways of the amygdala to subcortical destinations of the limbic system include the stria terminalis, which travels along the lateral aspect of the fornix and through the caudothalamic groove and terminates in the bed nucleus of the stria terminalis. The ventral amygdalofugal pathway is another important efferent pathway from the amygdala that originates from the BLA and CeA and connects to the striatum, namely to the nucleus accumbens, as well as to the basal forebrain, medial dorsal nucleus of the thalamus, and lateral hypothalamus[23]. As for the afferent pathways of the amygdala, fibers carrying inputs into the amygdala exhibit a considerable correspondence with efferent fibers carrying outputs from the amygdala. The amygdala receives information from all sensory inputs—which originate from the olfactory bulb and temporal/anterior cingulate cortices—and also receives visceral inputs, which are transmitted from the hypothalamus, septal area, orbital area, and parabrachial nucleus[22].

Here, we employed a restricted rabies virus system to provide a systematic dissection of whole-brain monosynaptic inputs onto CeA-CRF neurons. We found 44 regions throughout the brain that projected to CeA-CRF neurons, most of which were concentrated in the cortex, striatum, and thalamus. According to their identification and classification via fluorescent in situ hybridization (FISH), we revealed that these inputs were neurons mostly expressing $Ca^{2+}$/calmodulin-dependent protein kinase II (CaMKII), while a minority of inputs were neurons expressing glutamic acid decarboxylase (GAD1). Furthermore, by applying the CLARITY technique with 3D-reconstruction imaging, we unveiled a heterogeneous distribution and connectivity of CeA inputs deriving from different cortical regions. In addition, we registered our reconstructed neurons with the 3D mouse brain model from the Allen Brain Atlas, enabling a framework for visualizing the *en passant* brain regions where the connections with input fibers are located. Collectively, these data provide information on the comprehensive monosynaptic inputs onto CeA-CRF neurons at single-cell resolution, which may provide a morphological basis for a better understanding of the various roles of CeA-CRF neurons.

## Results

### Identification of monosynaptic inputs to CeA-CRF neurons via rabies-based retrograde viral tracing.
CeA-CRF neurons were genetically targeted based on CRF-Cre mice, which is a transgenic mouse line expressing Cre recombinase in CRF neurons[24]. In order to achieve cell-type-specific tracing, a genetically restricted two-virus approach was applied to identify the whole-brain monosynaptic inputs onto CRF neurons located in the CeA, which has been used widely to characterize presynaptic inputs with high efficiency and accuracy[25]. By injecting two Cre-dependent adeno-associated viruses (Fig. 1a), avian sarcoma leucosis virus envelope protein (TVA) and rabies glycoprotein G (RG) were simultaneously expressed in CeA-CRF neurons of CRF-Cre mice (Fig. 1c). Three weeks after the first helper-virus injection (Fig. 1b), a genetically modified rabies virus—which lacks the endogenous gene for glycoprotein G and has been modified to express DsRed and the avian virus envelope protein (Fig. 1a)—was delivered into the same region. The mice were perfused one week after rabies virus injection, allowing enough time for the rabies virus to retrogradely infect and express DsRed sufficiently in the input neurons (Fig. 1d). Starter neurons were identified by the coexpression of EGFP and DsRed around the injection site (Fig. 1e) and the input neurons were identified by the expression of DsRed that directly input to CeA-CRF neurons[26,27].

The specificity of our tracing approach was validated by several control experiments. To confirm the necessity of rabies glycoprotein, only AAV-DIO-TVA-EGFP was delivered to CRF-Cre mice before rabies injection, which resulted in neurons that coexpressed both EGFP and DsRed without extrinsic input only expressed as DsRed (Supplementary Fig. 1a). This result indicated that rabies glycoprotein was a key component for this rabies to retrogradely infect starter neurons. Similarly, only injecting AAV-DIO-RG and rabies into CRF-Cre mice did not result in any DsRed-labeled neurons (Supplementary Fig. 1b), indicating that TVA was necessary to infect starter neurons. Meanwhile, injecting AAV-DIO-RG, AAV-DIO-TVA-EGFP, and rabies into wild-type mice also did not result in any DsRed-labeled neurons (Supplementary Fig. 1c), indicating that all helper viruses expressed all components strictly depending on the expression of Cre recombinase. To further confirm the specificity

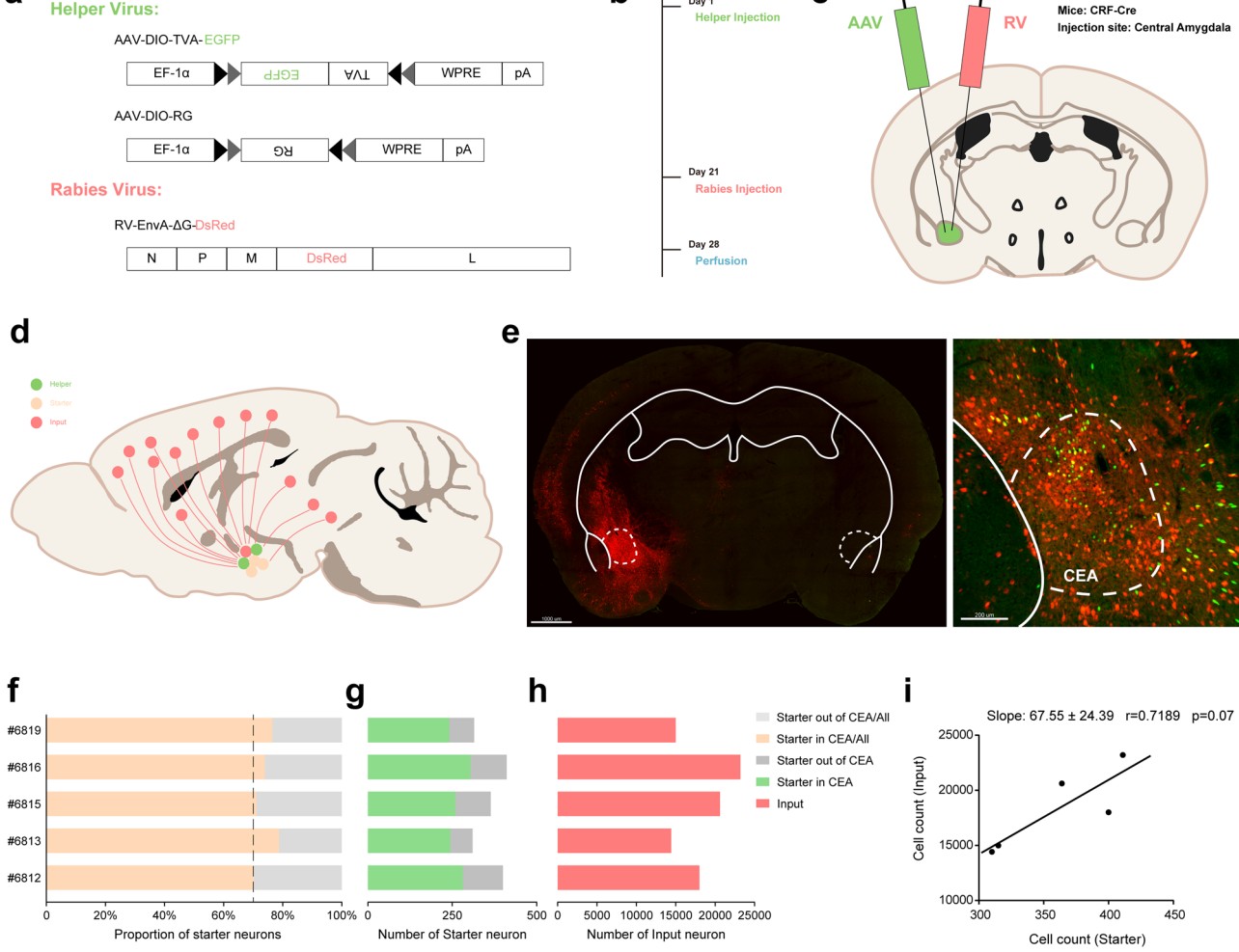

**Fig. 1 Identification of monosynaptic inputs to CeA-CRF neurons via rabies-based retrograde viral tracing. a** AAV helper viruses with Cre-dependent expression of the receptor of avian sarcoma leucosis virus envelope protein (TVA) and rabies glycoprotein G (RG) were used. Genetically modified rabies virus was pseudotyped with EnvA. The RG gene was replaced by EGFP. **b** Experimental timeline. AAV helper viruses were injected at day 1, and the rabies virus was injected at day 21 for the proper expression of the helper viruses. The infected mice were sacrificed on day 28. **c** Combination of the two-virus system and transgenic CRF-Cre mice allowed for brain-wide labeling of monosynaptic inputs (red) to CRF neurons (green) in the CeA. **d** Schematic of the input atlas of the CeA-CRF neurons at a whole-brain scale. **e** Tile-scan image shows the starter neurons in the CeA (scale bar: 1000 μm). Starter neurons were identified by the colocalization of EGFP (green) and DsRed (red) fluorescent proteins in the enlarged image (scale bar: 200 μm). **f** Proportion of starter neurons in the CeA. Only samples with more than 70% of the starters distributed within the CeA were used for further analysis. **g** Total number of starter neurons inside and outside of the CeA. **h** Total number of input neurons in the whole brain. **i** There was a linear relationship between starter and input neurons across all samples, indicating that the virus had the same transsynaptic efficiency in different samples. (N = 5).

of the neuronal types of starter neurons, we performed FISH. Most of the starter neurons were positive for the CRF mRNA probe (Supplementary Fig. 1d), which indicated that the starter neurons were CRF-expressing neurons and that the CRF-Cre line that we used had a high specificity. By injecting an AAV expressing EGFP driven by a CaMKII promoter in the secondary motor area, where the CaMKII-expressing input is mainly distributed, the fiber terminals were clearly observed in CeA (Supplementary Fig. 1e), which demonstrated the confirmation of the distant input regions and the fidelity of this rabies tracing system.

To verify the reliability of the approach in every tracing case, the whole brain of each CRF-Cre mouse in the virus-tracing group was sectioned to examine the distribution of starter neurons in the amygdala (Supplementary Fig. 2a). The distribution of starter neurons around the CeA was dissected (Fig. 1f), and the cases were counted and further analyzed only if the proportion of the starter neurons in the CeA exceeded 70% of the total starter neurons. The variability of the tracing approach

among different mice was further detected by counting the number of starter and input neurons (Fig. 1g, h). Moreover, the number of starter neurons was counted in every slice and plotted according to bregma coordinates. Most of the starter neurons (67.87 ± 3.31%) were located between bregma −0.8 to −1.4, where the CeA is located (Supplementary Fig. 2b). These data demonstrated that the injection site was relatively accurate in each case. Furthermore, regression analysis between starter and input neurons was conducted (Fig. 1i), and the quantified number of input neurons had a linear relationship with the number of starter neurons ($F$ (1,3) = 7.673), which indicated that the set of viruses exhibited a consistent infection efficiency across samples and that our retrograde tracing method was fairly stable.

**Quantitative analysis of brain-wide monosynaptic inputs to CeA-CRF neurons.** To dissect the whole-brain monosynaptic inputs to CeA-CRF neurons, the entire brain was coronally

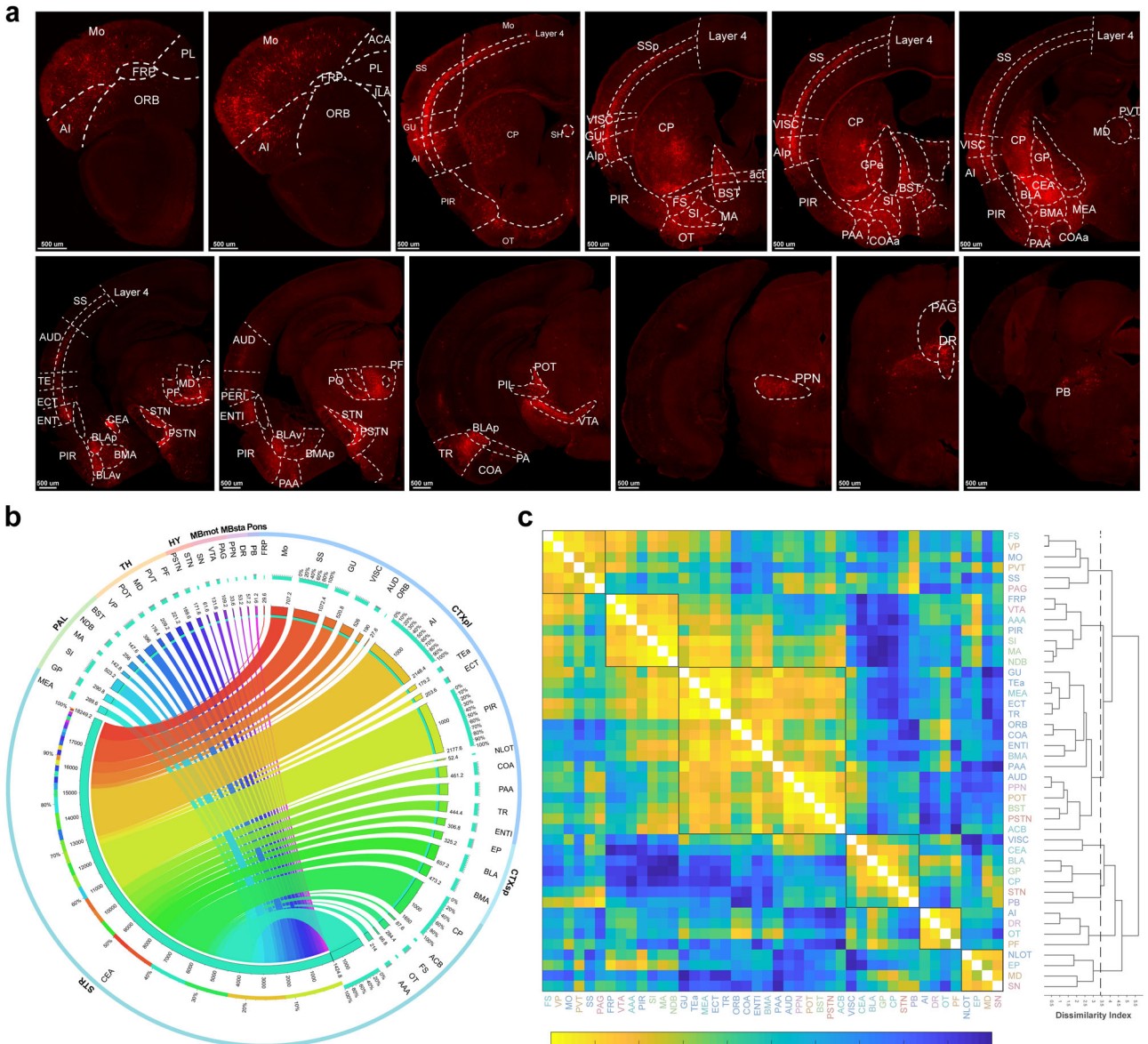

**Fig. 2 There are 44 different subregions that are clustered into six groups with direct input to CeA-CRF neurons. a** Representative coronal sections indicating the labeling of monosynaptic inputs and their nested subregions throughout the brain (scale bar: 500 μm). The anatomical abbreviations are shown in Supplementary Tables 1–3. **b** There are three circles in the Circos map that visualized the connection between the input regions and CeA. The inner layer shows the number of input neurons in 44 anatomical subregions, the middle layer shows the proportion of inputs in each subregion to the total inputs, and the outer layer shows the brain regions in which the subregions reside. The lines in the middle connect the input regions and start region (CeA), and the thickness of each line denotes the proportion of inputs from different brain regions. **c** The 44 subregions were clustered into six groups according to spearman correlations and hierarchal clustering (average method), and the font colors represent the brain regions in which these input neurons reside. These data suggest that brain regions within the same group play similar roles in specific behaviors and that their proportions may be positively correlated with the numbers of their inputs.

sectioned and reconstructed after imaging (Supplementary Fig. 3a, Supplementary Movie 1). The DsRed-labeled input neurons of every other slice were counted (Supplementary Fig. 2c). The representative coronal images showed that the monosynaptic inputs to CeA-CRF neurons were distributed throughout the whole brain (Fig. 2a), and most of the inputs were located in the subregions of the forebrain (Supplementary Fig. 3b).

Whole-brain quantified classification of the inputs enabled statistical analysis of anatomical specializations. By analyzing the distribution of inputs in specific subregions and the larger anatomical regions in which these subregions resided, a detailed projection-preference map to CeA-CRF neurons was generated (Fig. 2b).

Furthermore, by analyzing the correlation between each subregion in every sample, all 44 input subregions were clustered into six groups (Fig. 2c), which suggests that CeA-CRF neurons may be involved in the regulation of several behavioral processes via coordination with these highly distributed inputs across the strongly correlated brain regions[28]. In contrast, brain regions with fewer inputs may simply provide background information to the CeA (i.e., maintaining basic information exchange between these two brain regions[29]).

**Classification of brain-wide input patterns to CeA-CRF neurons.** To establish a refined map of the input atlas that demonstrates

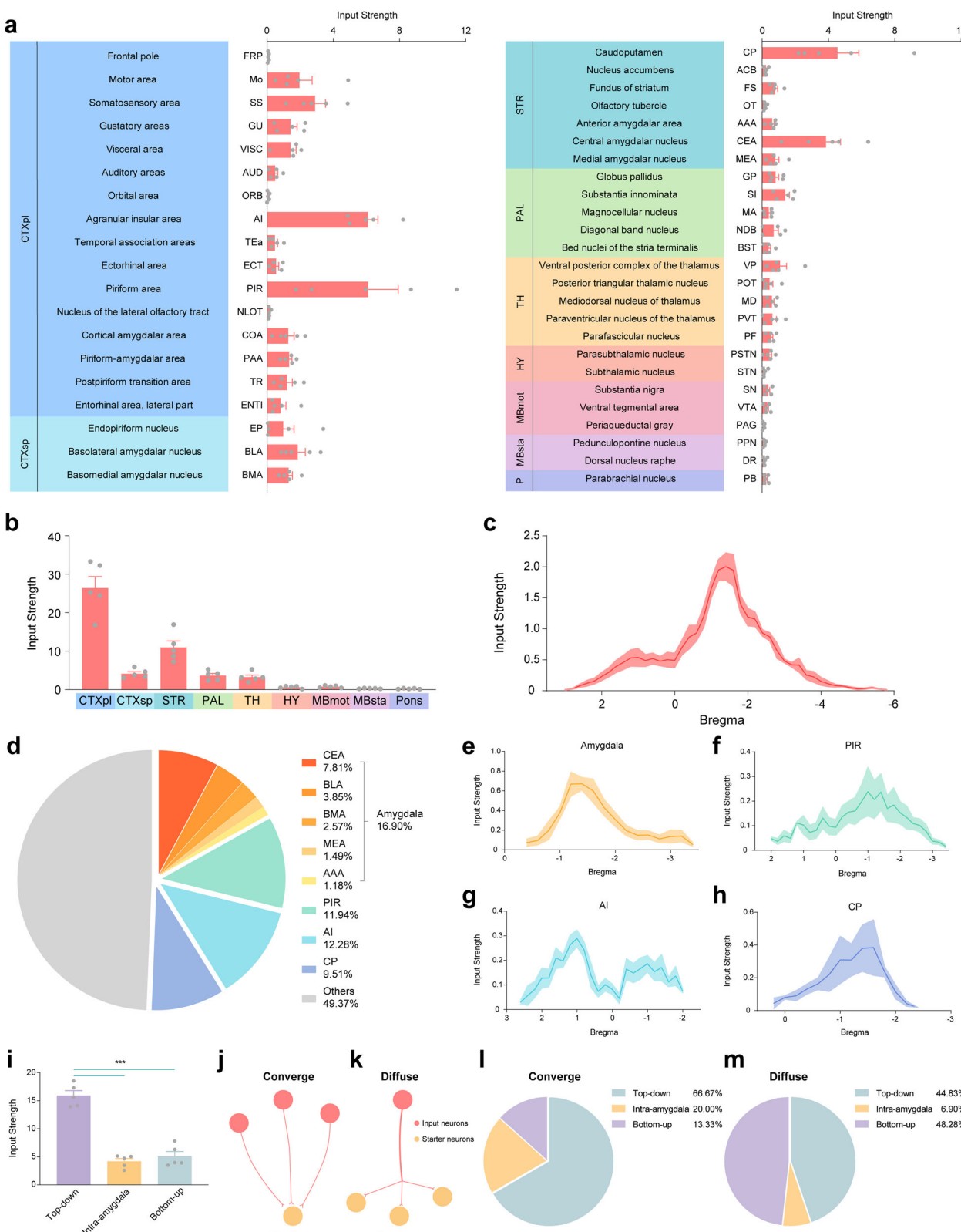

differences in input strengths between regions, the number of inputs was divided by the number of starters. Using this method, the input strengths of all 44 input subregions were recalculated and grouped in nine anatomical regions (Fig. 3a). More than half of the input subregions were in the cortical plate and striatum (23 out of 44, 52.27%). Consistent with this result, the input strength of the cortical plate was highest in these nine larger anatomical regions,

and the input strengths of the cortical plate and striatum were nearly three-quarters of the sum of all regions (74.13 ± 1.74%, Fig. 3b). Moreover, we found that the inputs were distributed from bregma +3.0 to −5.6 (Fig. 3c). Most of the input strength (73.54 ± 2.96%) was distributed between bregma 0 to −3.0, and the maximum value (2.00 ± 0.22) was at bregma −1.4, where the amygdala is located.

**Fig. 3 The whole-brain inputs of CeA-CRF neurons have a centralized distribution pattern. a** The input strength was defined as the number of input neurons divided by the number of starter neurons to normalize the variation in the number of starters between each case. The input strengths and variabilities in 44 nested anatomical subregions grouped by gross anatomical regions are shown. Notably, more than half of the input subregions were in the cortical plate (CTXpl) and Striatum (STR, 23 out of 44, 52.27%), and the input strengths in the AI, PIR, CP, and CeA were higher than those of other subregions. **b** The input strengths in each anatomical region and their variability are shown. The input strengths of CTXpl and STR were nearly three-quarters of the sum of all regions (74.13 ± 1.74%). **c** The input strengths (thick line) and the corresponding s.e.m (shadow) of inputs from anterior-to-posterior throughout the brain are shown. The inputs were centrally distributed, such that most of the input strength (73.54 ± 2.96%) was situated between bregma 0 and 3.0, and the maximum value of input strength was at bregma −1.4 (2.00 ± 0.22), where the amygdala is located. These results suggest that the axonal innervation of input neurons onto CeA-CRF neurons was mainly concentrated around the amygdala, and that it was gradually weakened by this boundary. **d** The percentages of input strengths in several subregions, including the amygdala (16.90 ± 2.41%), PIR (11.94 ± 3.63%), AI (12.28 ± 1.55%), and CP (9.51 ± 3.04%). The combined percentage of input strengths from these regions exceeded 50%. **e** The inputs located in the amygdala were distributed from bregma −0.4 to −3.4, and the maximum input strength was at bregma −1.4 (0.67 ± 0.07%). **f** The distribution of inputs in the PIR ranged from bregma 2.0 to −3.4, and the maximum input strength was at bregma −1.4 (0.24 ± 0.10%). **g** The inputs in the AI were distributed from bregma 2.6 to −2.0, but there were two peaks at bregma 1.0 (0.29 ± 0.03%) and −1.0 (0.19 ± 0.03%). **h** In the CP subregion, inputs were distributed from bregma 0.2 to −2.4, and bregma −1.6 was the location of maximum input strength (0.38 ± 0.17%). The line refers to the mean input strength, and the shadow refers to the corresponding s.e.m. (**e–h**). **i** The input regions was grouped into the top–down, intra-amygdala, and bottom–up groups according to the relative positions between the input regions and the start region, CeA (the grouping of subregions is detailed in Supplementary Table 1–3). The input strength in the top–down group was significantly larger than that of the other two regions. ($F$ (2, 12) = 72.88, $p < 0.0001$, one-way ANOVA with Tukey correction). **j** The input strength was defined as the number of input neurons divided by the number of starter neurons, which also represents the input patterns of these brain regions. The brain regions with input strengths greater than a value of 1.0 were designated to the convergent group, which indicated that more than one input neuron in these regions may innervate a single starter neuron in the CeA. **k** On the contrary, the brain regions in the diffuse group were defined as those with an input strength value of <1.0, which demonstrated that an individual input neuron in these regions may innervate multiple starter neurons in the CeA. **l** Most of the inputs in top–down regions converged onto a single starter neuron in the CeA. The percentages of convergent input patterns in the subregions of top–down (66.67%), intra-amygdala (20.00%,) and bottom–up (13.33%) groups are shown. **m** In contrast, the inputs in bottom–up regions diffused into more than one starter neuron in the CeA. The percentages of diffuse input patterns in the subregions of top–down (44.83%), intra-amygdala (6.90%), and bottom–up (48.28%) groups are shown. Data are mean ± s.e.m., $N = 5$.

Investigation on the input regions revealed that there were four major regions that contributed half (50.63 ± 3.98%) of the input strength onto CeA-CRF neurons, namely the piriform area (11.94 ± 3.63%), agranular insular area (12.28 ± 1.55%), caudoputamen (9.51 ± 3.04%), and amygdala (16.90 ± 2.41%, Fig. 3d). The inputs located in the amygdala were distributed from bregma −0.4 to −3.4, and the maximum input strength was at bregma −1.4 (0.67 ± 0.07%, Fig. 3e). The distribution of inputs in the piriform area was more extensive, ranging from bregma 2.0 to −3.4, and the maximum input strength was at bregma −1.4 (0.24 ± 0.10%, Fig. 3f). The inputs in the agranular insular area were distributed from bregma 2.6 to −2.0, but there were two peaks of its input strength, which were at bregma 1.0 (0.29 ± 0.03%) and −1.0 (0.19 ± 0.03%, Fig. 3g). Finally, in caudoputamen, inputs were distributed from bregma 0.2 to −2.4, and bregma −1.6 was the location of maximum input strength (0.38 ± 0.17%, Fig. 3h). These results suggest that the input pattern of whole-brain projections to CeA-CRF neurons has a centralized distribution quality, as evidenced by the fact that the axonal innervation of input neurons onto CeA-CRF neurons was mainly concentrated in the brain nuclei around the amygdala, and that it was gradually weakened by this boundary.

Based on the distribution pattern of inputs, we divided them into three groups: (1) the intra-amygdala group, in which inputs were located in subregions of the amygdala (e.g., CeA) (Supplementary Table 1); (2) the top–down group, in which the main inputs resided anteriorly to the amygdala (e.g., frontal pole[30]) (Supplementary Table 2); and (3) the bottom–up group, in which the main inputs were situated posteriorly to the amygdala (e.g., periaqueductal gray[31]) (Supplementary Table 3). We found that the top–down input strength was remarkably stronger than that of the input strength from the intra-amygdala and bottom–up groups ($F$ (2, 12) = 72.88, $p < 0.0001$, Fig. 3i), which indicated that CeA-CRF neurons mainly received information from top–down regions.

According to the input strength of each region, we divided these regions into two groups, which also represented the input

patterns of these brain regions. The brain regions with input strength greater than a value of 1.0 were designated to the convergent group, which indicated that more than one input neuron in these regions innervated a single starter neuron, and that information processed by multiple input neurons in these regions converged onto single CeA-CRF starter neurons (Fig. 3j). On the contrary, an individual input neuron in the brain regions with an input strength value <1.0 could probably innervate multiple starter neurons, and these groups that may diffusely pass information into starter CeA-CRF neurons were defined as diffuse groups under these circumstances (Fig. 3k). Upon examining the compositions of these two groups, the majority of the convergent group (66.67%) consisted of brain regions from the top–down group (Fig. 3l), whereas the majority of the diffuse group (48.28%) consisted of brain regions from the bottom–up group (Fig. 3m). Taken together, for the brain-wide, the top–down inputs tended to converge, whereas the bottom–up inputs tended to diverge, onto CeA-CRF neurons.

**Neuron-type characterization of brain-wide monosynaptic inputs to CeA-CRF neurons.** To further classify the inputs and identify the information that they transferred, the inputs were characterized by two markers. One marker is CaMKII, which is also a primary marker of excitatory neurons in the cortex[32]. The other marker is GAD1, which is also a primary marker of most inhibitory neurons[33]. Using the CaMKII FISH probe (Supplementary Fig. 4a) and GAD1 FISH probe (Supplementary Fig. 4b) respectively (Supplementary Table 4), most of the input neurons in the four primary regions (piriform area: 52.55 ± 1.91%, agranular insular area: 69.77 ± 2.07%, caudoputamen: 51.36 ± 0.67%, CeA: 52.18 ± 2.74%) were positive for the CaMKII mRNA probe, whereas only small percentages of input neurons (piriform area: 21.70 ± 1.06%, agranular insular area: 18.64 ± 0.05%, caudoputamen: 23.34 ± 0.64%, CeA: 41.57 ± 0.47%) were GAD1-positive (Fig. 4a). In the majority of these input subregions—except for the orbital area, the nucleus of the lateral olfactory tract, diagonal band nucleus, a ventral posterior complex of the thalamus,

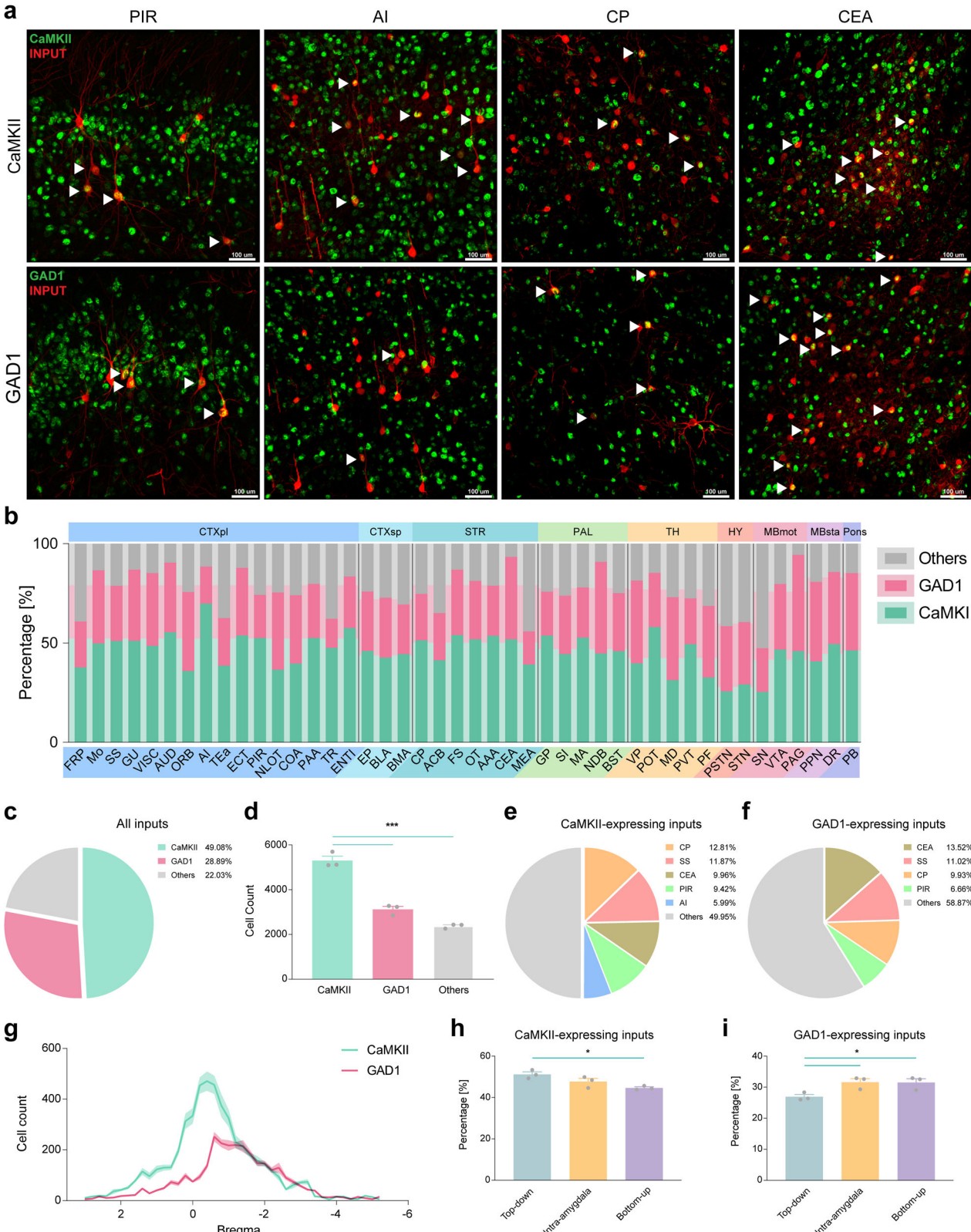

mediodorsal nucleus of the thalamus, parafascicular nucleus, parasubthalamic nucleus, subthalamic nucleus, and periaqueductal gray—there were more inputs expressing CaMKII than those expressing GAD1. In addition, only the hypothalamus had more inputs expressing GAD1 than those expressing CaMKII (Fig. 4b). Overall, most of the inputs were expressing CaMKII (49.09 ± 1.05%), the percentage of which was significantly higher

than that of inputs expressing GAD1 (28.89 ± 0.80%) and other types of inputs (22.03 ± 0.95%, $F$ (2, 6) = 115.8, $p$ = 0.0001, Fig. 4c, d). CaMKII-expressing input neurons were mostly distributed in the caudoputamen (12.81 ± 0.35%), somatosensory area (11.87 ± 0.37%), CeA (9.96 ± 0.21%), piriform area (9.43 ± 0.20%), and agranular insular area (5.99 ± 0.19%, Fig. 4e). In contrast, GAD1-expressing input neurons were mainly located

**Fig. 4 Top–down inputs are CaMKII-expressing, while bottom–up inputs are mostly GAD1-expressing. a** Representative image in the CeA, PIR, AI, and CP demonstrating neuronal subtypes of input neurons (red) identified by CaMKII or GAD1 FISH probes (green) (scale bar: 100 μm). **b** The percentages of CaMKII- and GAD1-expressing inputs in 44 nested subregions (front) and their anatomical regions (background). In all these input subregions—except for the ORB, NLOT, NDB, VP, MD, PF, PSTN, STN, and PAG—there were more inputs that expressed CaMKII than those that expressed GAD1. Additionally, only the hypothalamus had more inputs expressing GAD1 than those expressing CaMKII. **c** Nearly half of the inputs expressed CaMKII (49.09 ± 1.05%), while less than one-third of the inputs expressed GAD1 (28.89 ± 0.80%). **d** The number of CaMKII-expressing inputs was markedly higher than that of GAD1-expressing inputs ($F (2, 6) = 115.8$, $p < 0.0001$). **e** Nearly 50% of CaMKII-expressing inputs were located in the CP (12.81 ± 0.35%), SS (11.87 ± 0.37%), CeA (9.96 ± 0.21%), PIR (9.43 ± 0.20%), and AI (5.99 ± 0.19%). **f** While the major brain regions of GAD1-expressing inputs included the CeA (13.52 ± 0.24%), SS (11.02 ± 1.19%), CP (9.92 ± 0.69%), and PIR (6.66 ± 0.63%). Furthermore, these expression data suggest that the CeA, SS, and PIR may play both excitatory and inhibitory roles in CeA-CRF-related behaviors. **g** The line refers to the mean number of CaMKII- and GAD1-expressing inputs and the shadow refers to the corresponding s.e.m. from anterior-to-posterior, throughout the brain are shown. The distribution patterns of inputs for these two major neuronal types were relatively similar, but there were more inputs expressing CaMKII in the first half of this spatial distribution than those expressing GAD1, whereas in the second half the numbers of these two populations were similar. **h** The percentage of CaMKII-expressing neurons in the top–down group was significantly larger than that in the bottom–up group ($F [2, 6] = 7.867$, $p = 0.0210$). **i** The percentage of GAD1-expressing neurons in the top–down group was markedly smaller than that in the other two groups ($F [2, 6] = 6.458$, $p = 0.0319$). These results indicated that the distribution of CaMKII- and GAD1-expressing inputs were also concentrated, consistent with the previous results. In addition, the top–down inputs in the cortical regions may convey mostly excitatory information, while bottom–up inputs were mostly inhibitory. Data are mean ± s.e.m., $N = 3$, one-way ANOVA with Tukey correction.

in the CeA (13.52 ± 0.24%), somatosensory area (11.02 ± 1.19%), caudoputamen (9.92 ± 0.69%), and piriform area (6.66 ± 0.63%, Fig. 4f). These results suggest that most of the information that CeA-CRF neurons received from cortical areas was excitatory, and that somatosensory area and piriform area may have both excitatory and inhibitory inputs onto CeA-CRF neurons, suggesting bidirectional modulation of some behaviors.

Next, we investigated the anterior-to-posterior distribution patterns of both CaMKII- and GAD1-expressing inputs throughout the brain. Each of the distribution curves of CaMKII- and GAD1-expressing inputs had only one peak; the maximum number of CaMKII-expressing inputs was distributed at bregma −0.4 (471.33 ± 38.12), whereas that of GAD1-expressing inputs was at bregma −0.6 (252.33 ± 18.48), which was consistent with the distribution of overall inputs. In the first half of the curve, there were remarkably more input neurons expressing CaMKII than those expressing GAD1, while in the second half, the numbers of these two neurons were similar (Fig. 4g).

By using the grouping rules of top–down, intra-amygdala, and bottom–up groups, the numbers of both CaMKII- ($F [2, 6] = 313.6$, $p < 0.001$) and GAD1-expressing inputs ($F [2, 6] = 102.7$, $p < 0.001$) in the top–down group were significantly higher than those in the intra-amygdala group and bottom–up group (Supplementary Fig. 4c–f). The majority of inputs in these three groups were expressing CaMKII (Supplementary Fig. 4g–l), and the percentage of CaMKII-expressing inputs in the top–down group was significantly higher than that in the bottom–up group ($F [2, 6] = 7.867$, $p = 0.02$, Fig. 4h). In contrast, the percentage of GAD1-expressing inputs in the top–down group was significantly lower than that in the other two groups ($F [2, 6] = 6.458$, $p = 0.03$, Fig. 4i). These data indicate that the CaMKII-expressing input neurons in the cortical regions of the top–down group may contribute the most excitatory inputs onto CeA-CRF neurons, while the input neurons in the bottom–up group were mainly expressing GAD1 and may provide primarily inhibitory inputs onto CeA-CRF neurons.

**Reconstruction and morphologic analysis of inputs in the cortices of SS, AI, and PIR.** The sizes and shapes of dendrites and axons play decisive roles in neuronal information processing[34]. Across the coronal sections of the somatosensory area, agranular insular area, and piriform area, which were three major input subregions deriving from cortical areas, the inputs located in these areas exhibited specific distribution and connectivity patterns (Fig. 5a). To investigate the relationships of inputs between the

layers in these three brain regions, 48 input neurons were reconstructed (Supplementary Fig. 5), and 11 morphological characteristics were extracted by L-measure (Supplementary Table 5). By examining correlations of the morphological features between these input neurons, we found that they were mainly clustered into three groups, which had significant differences in terms of soma surface ($F [2, 45] = 7.635$, $p = 0.0014$, Supplementary Fig. 6a), number of stems ($F [2, 45] = 3.550$, $p = 0.0370$, Supplementary Fig. 6b), bifurcations ($F [2, 45] = 63.89$, $p < 0.001$, Supplementary Fig. 6c), branches ($F [2, 45] = 59.55$, $p < 0.001$, Supplementary Fig. 6d), tips ($F [2, 45] = 51.45$, $p < 0.001$, Supplementary Fig. 6e), depth ($F [2, 45] = 14.99$, $P < 0.001$, Supplementary Fig. 6g), depth/width ratio ($F [2, 45] = 12.75$, $p < 0.001$, Supplementary Fig. 6h) and branch path length ($F [2, 45] = 32.99$, $p < 0.001$, Supplementary Fig. 6k). Representative neurons shown in each group indicated that the complexity of neuronal structure occupied a larger proportion of the clustering weight (Fig. 5b). In addition, neurons in the agranular insular area were mainly clustered in group 1 (12 out of 14), while neurons in the somatosensory area and piriform area were more evenly distributed among the three groups. These results of morphological characterizations indicate that these three clusters of inputs mainly differed in terms of the geometry complexities of their processes, whereas there were no obvious differences in the sizes of the somata or processes of these neurons.

Next, correlations among these 11 morphological indicators and somatic depths were analyzed (Fig. 5c). There were strong positive correlations in terms of the numbers of stems, bifurcations, branches, and tips, all of which are indicators of the complexity of neuronal processes. However, we found that the complexity of neuronal processes had no correlation with the overall size of neurons, the latter of which was described by the width, height, and depth morphological parameters. Furthermore, by analyzing the morphological indicators with respect to somatic depth, it was revealed that there were negative correlations between somatic depth and the numbers of stems ($r^2 = 0.1080$, $p = 0.0226$, Fig. 5d), bifurcations ($r^2 = 0.2812$, $p = 0.0001$, Fig. 5e), branches ($r^2 = 0.2922$, $p < 0.0001$, Fig. 5f), tips ($r^2 = 0.2923$, $p < 0.0001$, Fig. 5g), and branch path length ($r^2 = 0.1920$, $p = 0.0018$, Fig. 5h), suggesting that the cellular architecture of input neurons had a tendency for cortical depth-specific organization, such that deeper input neurons distributed along the depth axis were larger but had simpler structures.

By immunofluorescent labeling of NECAB1, which is a marker of layer 4 in the neocortex, it was found that the inputs were densely distributed in layer 4 on the dorsal side of the

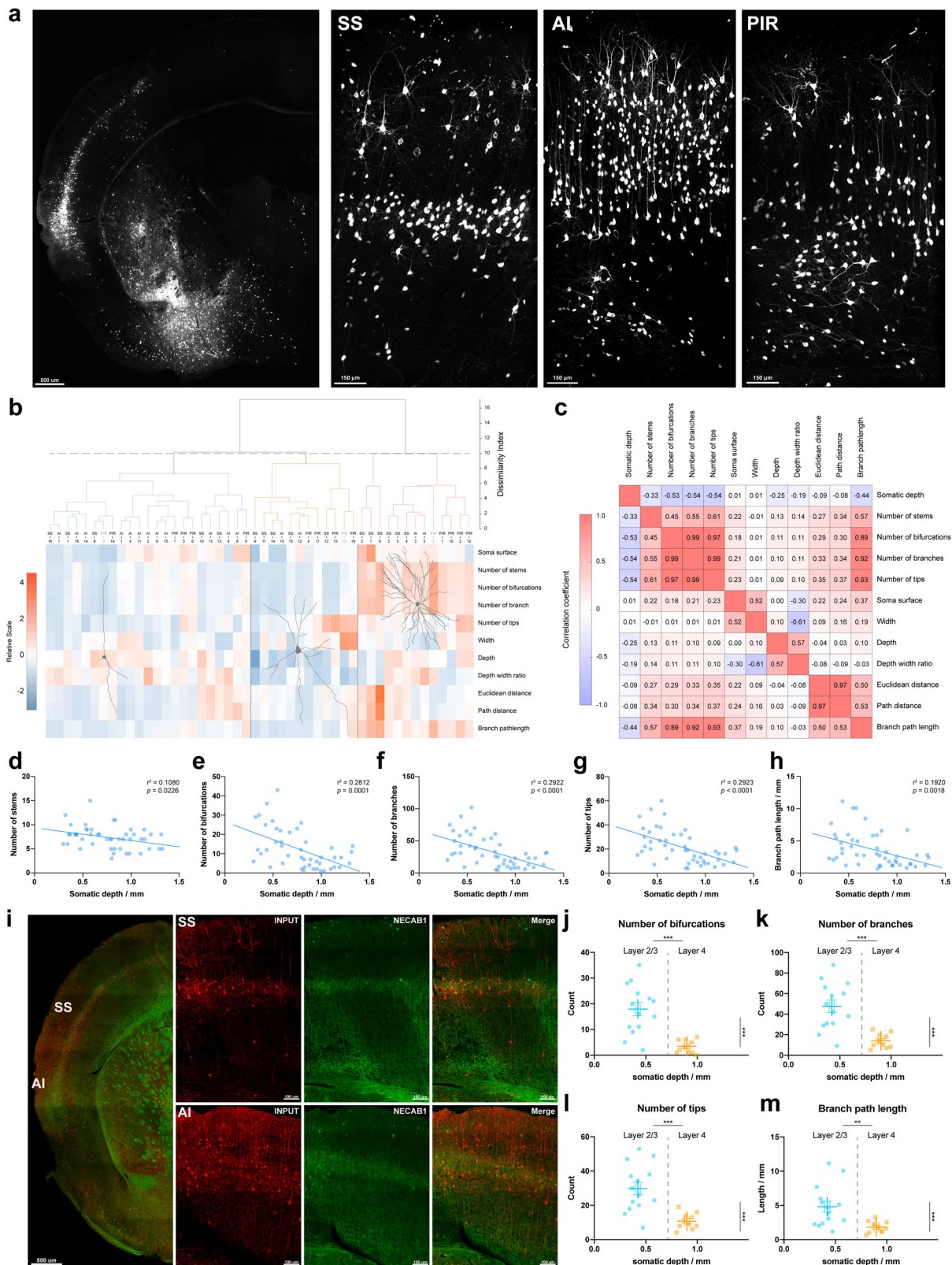

somatosensory area, whereas they were more dispersed toward the ventral side. In the agranular insular area, there was a higher distribution density of inputs in layer 4, and they also had a wider distribution in other layers; in contrast, this specific pattern was absent on the ventral side of the section, where the piriform area was located (Fig. 5i). In the layer-specific inputs in the somatosensory area and agranular insular area, the numbers of

bifurcations ($t$ [22] = 4.650, $p < 0.001$, Fig. 5j), branches ($t$ [22] = 4.487, $p < 0.001$, Fig. 5k), tips ($t$ [22] = 4.223, $p < 0.001$, Fig. 5l) and branch path length ($t$ [22] = 3.036, $p = 0.0061$, Fig. 5m) in layer 2/3 were remarkably larger than those in layer 4, which was consistent with our previous results showing that the numbers of bifurcations, branches, tips, and branch path length were negative correlated with somatic depth.

**Fig. 5 Inputs in the cortices of SS, AI, and PIR have a cortical layer-specific property and are clustered into three groups according to their morphological characteristics. a** Representative image of tile-scanned coronal sections of inputs in the SS, AI, and PIR. (Scale bar: 500 μm and 150 μm, respectively). **b** All reconstructed neurons were clustered into three groups by their 11 morphological characteristics. The representative neurons displayed in each group illustrated that the complexity of neuronal structure occupied a larger proportion in the clustering weight and that there were no significant differences in the sizes of neuronal somata or their fibers. The relative scale data were normalized by the $z$ score method for each morphological parameter. **c** Correlation between 11 morphological parameters and somatic depth. Strong correlations were found between parameters demonstrating the structural complexity of neurons (i.e., number of stems, bifurcations, branches, and tips), while parameters showing the sizes of neurons also exhibited strong correlations (i.e., Euclidean distance, path distance, and branch path length). **d–h** The complexity of the inputs—numbers of stems (**d**, $r^2 = 0.1080$, $p = 0.0226$), bifurcations (**e**, $r^2 = 0.2812$, $p = 0.0001$), branches (**f**, $r^2 = 0.2922$, $p < 0.0001$), tips (**g**, $r^2 = 0.2923$, $p < 0.0001$), and branch path length (**h**, $r^2 = 0.1920$, $p = 0.0018$)—in these three regions were negatively correlated with their somatic depths, suggesting that the cytoarchitecture of the input neurons tended to have cortical depth-specific organization, and that inputs with deeper distributions along the depth axis were larger but simpler in structure ($N = 48$). **i** Immunofluorescent-labeled NECAB1 (green), a marker of layer 4, illustrating that inputs (red) were densely distributed in layer 4 on the dorsal side of the SS, while they were more dispersed toward the ventral side in the cortex (Scale bar: 500 μm and 100 μm, respectively). **j–m** The numbers of bifurcations (**j**, $t$ [22] = 4.650, $p < 0.001$), branches (**k**, $t$ [22] = 4.487, $p < 0.001$), tips (**l**, $t$ [22] = 4.223, $p < 0.001$) and branch path length (**m**, $t$ [22] = 3.036, $p = 0.0061$) of the input neurons in layer 2/3 were markedly greater than those in layer 4 of SS and AI with respect to layers and somatic depth. Data are mean ± s.e.m., two-tailed unpaired $t$ test, layer 2/3: $N = 14$, layer 4: $N = 10$.

As expected, the somatic depth ($t$ [22] = 13.79, $p < 0.001$) of input neurons in layer 2/3 was notably smaller than that of input neurons in layer 4. However, there was no significant difference in the number of stems ($t$ [22] = 1.846, $p = 0.0783$) between these two layers. These results suggested that the inputs in layer 2/3 had more complex branching patterns—which is a strong indicator of dendritic complexity—compared to those in layer 4.

**Pathway registration of reconstructed inputs from the SS, MD, and PAG reveals the connections in their *en passant* structures.** To further investigate the *en passant* structures of the long-range projection fibers of inputs, three representative input neurons in the somatosensory area, mediodorsal nucleus of thalamus, and periaqueductal gray was completely reconstructed in the light-sheet images at the micron scale (Supplementary Fig. 7a). To accurately visualize the paths of these long-range projection fibers of the CeA-CRF inputs throughout the mouse brain, a standardized 3D brain model was created from the mouse brain reference atlas of the Allen Institute. In this framework, these three reconstructed input neurons were registered in the mouse brain model (Supplementary Fig. 7b). Moreover, by loading the models of brain regions through which the long-range projection fibers passed, a detailed anatomical projecting pathway was revealed.

The projecting axon of the input neuron in the somatosensory area started from the cortex, then directly passed through three regions of white matter—namely the supra-callosal cerebral white matter (scwm), auditory radiation (ar), and corpus callosum (cc)—and arrived at the caudoputamen, after which it exited the bottom of the caudoputamen and ultimately innervated the amygdala (Supplementary Fig. 7c, Supplementary Movie 2). The traced input neuron in the mediodorsal nucleus of thalamus sent out its axon to the paracentral nucleus (PCN) via the crossed tectospinal pathway (tspc), passed through the ventral medial nucleus of the thalamus (VM) to the zona incerta (ZI) of hypothalamus, and then turned into the pallidum. Finally, it reached the amygdala by passing through the optic tract (opt) (Supplementary Fig. 7d, Supplementary Movie 3). The input neuron in the periaqueductal gray had the longest axon among these three reconstructed neurons (Fig. 6a). After exiting the periaqueductal gray, the axon of this neuron entered the subparafascicular nucleus (SPF) of the thalamus through the midbrain reticular nucleus (MRN), passed through the medial lemniscus (ml) to the zona incerta and subthalamic nucleus of the hypothalamus, and finally entered the amygdala through the GP and caudoputamen (Fig. 6b, Supplementary Movie 4). These results highlight the complexity of the long-range projections of CeA-CRF input neurons and illustrate every specific structure

through which input neurons in different brain regions project to the CeA.

To investigate whether these long-range projecting fibers have connections in the passing structures, myelin basic protein (MBP), a protein marker of myelin that wraps axons[35], was labeled in these structures. However, MBP was not colocalized with some segments of the input fibers, which indicated that these fibers were not completely encapsulated in myelin (Supplementary Fig. 8a). Furthermore, postsynaptic density protein 95 (PSD-95), a pivotal postsynaptic scaffolding protein in excitatory neurons[36], was also labeled on the slices (Supplementary Fig. 8b). PSD-95 was colocalized with some segments of the input fibers in their *en passant* structures, which were the parts of the caudoputamen (Supplementary Fig. 8c) and the thalamus (Supplementary Fig. 8d). These results suggest that these fibers of input neurons have connections within these *en passant* structures, and these fiber-connecting neurons may, in turn, modulate the information transmitted from the inputs.

## Discussion
The amygdala is important for emotional and motivational processing, and its circuits and functions have been demonstrated to be evolutionarily conserved[37]. Hence, dissecting the neural circuitry and elucidating the input patterns of the amygdala in animal models—aside from improving our understanding of the intrinsic function of the amygdala—may help to further determine the relationship of amygdalar dysfunction with psychiatric disorders. Our present study comprehensively classified the whole-brain monosynaptic inputs that target CeA-CRF neurons at single-cell resolution. However, there are multiple neuronal subtypes with different molecular markers distributed in CeA[6]. In addition to CRF, a previous study has mapped the monosynaptic-input atlas of SST and PKC-δ at the whole-brain scale[38]. All three molecularly distinct neurons within CeA receive extensive monosynaptic information throughout the brain, with inputs distributed from the anterior olfactory areas to the posterior midbrain regions. However, in comparison, CeA-SST neurons receive more inputs from the cortex, CeA-PKC-δ neurons receive more inputs from the striatum, and CeA-CRF neurons receive their main inputs from both cortex and striatum.

To prevent any difference in the number of starter neurons between individuals from affecting the number of input neurons in brain regions, we normalized the number of inputs by dividing the number of starters to determine input strength. Through this processing, it was also found that there were two different input patterns of these subregions, namely convergent and diffuse patterns. In the convergent pattern, more than one input neuron

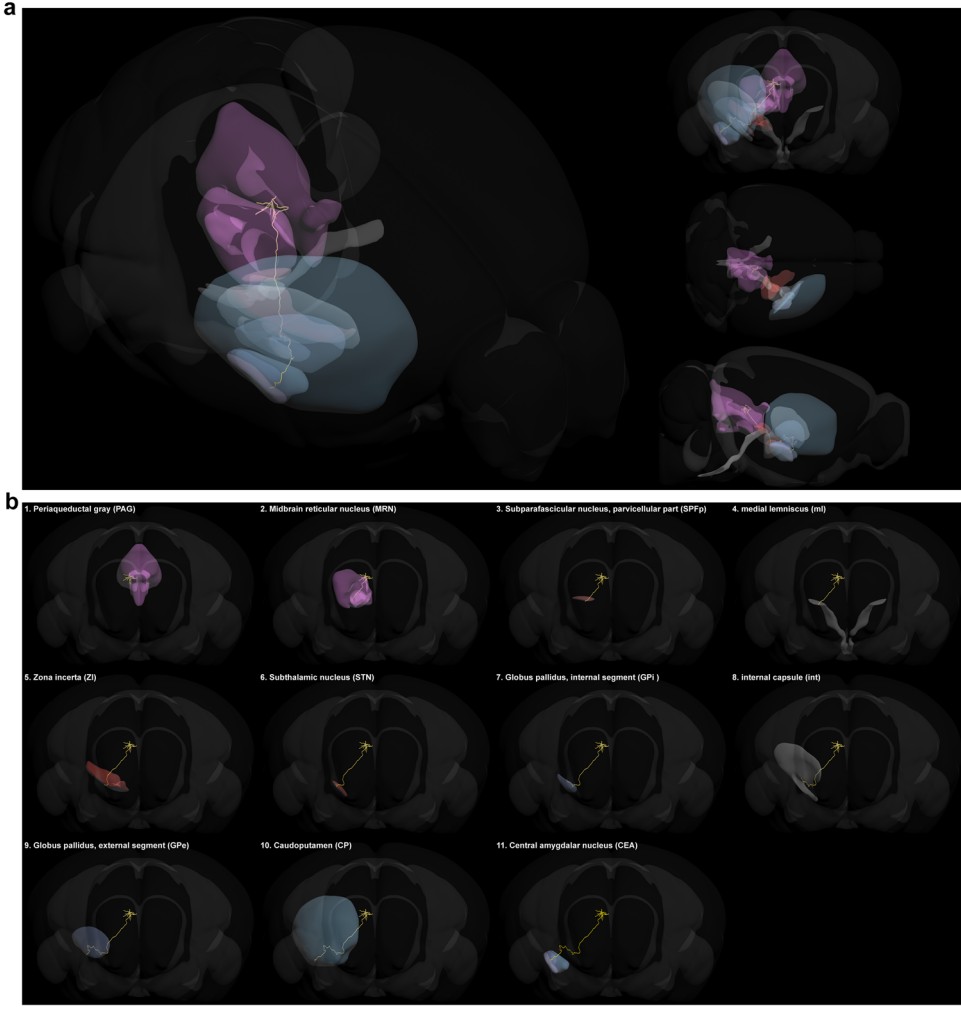

**Fig. 6 Registration of 3D reconstructed input neurons traced from the PAG in a standardized mouse brain reference atlas. a** A detailed pathway of an input neuron in the periaqueductal gray (PAG) is shown in the 3D-reference atlas with its three views. **b** The input neuron located in the PAG was the furthest from the CeA among the three input neurons that we reconstructed. This neuron passed through the PAG, midbrain reticular nucleus (MRN), subparafascicular nucleus (SPF), medial lemniscus (ml), zona incerta (ZI), subthalamic nucleus (STN), Globus pallidus, internal segment (GPi), internal capsule (int), Globus pallidus, external segment (GPe), and caudoputamen (CP) in turn before finally reaching the CeA.

in a given subregion innervated a single starter neuron. On the contrary, in the diffuse pattern, a single input neuron could probably project to multiple starter neurons. However, owing to the limitations of the methodology in this study, we cannot examine every single starter that the inputs innervate. More specifically, there may be some of the inputs in the diffuse group that only project to a fraction of the starters, whose number is even smaller than the inputs, resulting in a convergent pattern. Interestingly, similar convergent and diffuse input patterns have also been observed and well-studied in reward circuits[39]. In our present study, we found that the CeA received convergent information from top–down input regions, which were mainly distributed in various regions of the cortex (e.g., orbital area, somatosensory area, agranular insular area). These regions are known to process and transmit information related to sensations, emotions, and cognition, all of which become associated after their convergence in the amygdala, the processing of which then instructs the body to adaptively respond to external stimuli[3].

In our present study, CaMKII was used as a marker for excitatory neurons in the cortical regions, while inhibitory neurons may be identified by GAD1 expression. However, there is no single marker that can represent either all excitatory neurons or inhibitory neurons[40]. That said, CaMKII-positive neurons

account for over 70% of hippocampal pyramidal neurons and colocalize with most glutamatergic neurons in the neocortex and other brain regions. Notably, CaMKII is not a reliable marker of excitatory neurons in subcortical regions (Supplementary Fig. 4m). Meanwhile, GAD1-positive neurons mainly represent inhibitory interneurons[41]. Collectively, these known features account for CaMKII-positive neurons and GAD1-positive neurons in our present study not fully recapitulating all our labeled input neurons. Therefore, those other neurons may also be excitatory neurons that express Vglut but not CaMKII[42], and the CaMKII-expressing neurons in subcortical regions may be inhibitory neurons. Nevertheless, we found that GAD1-labeled GABAergic neurons at distance were all long-range-projecting neurons targeting CeA-CRF neurons. This finding is consistent with previous reports showing that GABAergic long-range-projecting neurons in the cortex preferentially target inhibitory interneurons in subcortical regions, which allows them to control remote target areas via disinhibition[43], such as amygdala[44]. And these GABAergic long-range-projection neurons have also been found in subcortical regions[45].

Based on the efficient tracing methods[46,47] and imaging[48,49], our results uncovered that these long-range projecting fibers were not completely wrapped by myelin. In addition, the colocalization

of the axons with PSD-95 provided further evidence that the fibers did not randomly cross these structures, but were associated with their *en passant* structures and received the information transmitted by them. Among all neurons we reconstructed, the neuron located in the supplemental somatosensory area had the shortest input pathway, directly crossing three white matters and reaching the CeA only through the caudoputamen (Supplementary Fig. 9a). This may be partly due to its relative proximity to the endpoint, and more likely because of the existence of a fast pathway for the transmission of emotional regulation information from supplemental somatosensory area neurons to CeA-CRF neurons[50], and caudoputamen may modulate this pathway by conveying sensorimotor information[51]. Similarly, the mediodorsal nucleus of the thalamus transmits cognitive memory-related information from the prefrontal cortex and other cortical structures[52], whereas its emitted fibers pass through the paracentral nucleus in the thalamus, which also receives input from cortical information and corrects the afferent information from the mediodorsal nucleus of thalamus[53] (Supplementary Fig. 9b). Finally, the reconstructed neuron in the periaqueductal gray is the farthest away from the CeA, and its fiber passes the most structures. The periaqueductal gray and MRN participate in many functions, and periaqueductal gray conveys information involved in motivated behavior and processes controlling not only aversive but also appetitive behavior[54]. MRN participates in autonomic, motor, sensory, cognitive, and mood-related functions[55]. However, when the fiber reaches the subparafascicular nucleus through the thalamus, the function becomes simple and concrete. The subparafascicular nucleus participates in sexually related and conditioned fear behaviors[56], which are both highly involved in the amygdala[57]. Like the previous two traced neurons, the fiber undergoes a series of modifications from several motion-related structures before finally entering CeA (Supplementary Fig. 9c).

Our present study systematically mapped the whole-brain input atlas onto CeA-CRF neurons and identified subtypes of input neurons. Furthermore, we characterized the morphological features of inputs distributed in the cortex and fully reconstructed the projection pathways of individual input neurons, and registered them with a 3D brain atlas, the latter of which enabled identification of *en passant* regions. Collectively, these results provide a structural foundation for subsequent functional studies interrogating the many physiological and behavioral roles of CeA-CRF neurons. In turn, this may provide insights into improving treatments for diseases and mental disorders related to CeA dysfunction.

## Methods

**Animals**. CRF-Cre mice were purchased from Jackson Laboratory (012704). All adult CRF-Cre male mice (8–16 weeks old) were group-housed at 3–4 mice per cage under a 12/12-h light/dark cycle (lights on 07:00 AM), at a room temperature of 22 °C and relative humidity of 50–60%. Food and water were provided ad libitum. All animal procedures were approved and conducted in accordance with the Institutional Animal Care and Use Committee at the University of Science and Technology of China. All efforts were made to minimize animal suffering as well as the number of animals used.

**Surgical procedures**. For surgeries, each mouse was deeply anesthetized with sodium pentobarbital (40 mg/kg, intraperitoneal injection) and fixed in a stereotaxic head holder (Stoelting Co., 51925). Eye ointment was used to prevent dry eyes during surgery. The hair on top of each mouse's head was shaved with a clipper to expose the scalp, which was then washed with double-distilled water and sterilized with 75% (v/v) ethanol to prevent inflammation. Subsequently, a sagittal incision was made with sterile scissors and forceps to expose the skull. We adjusted the head holder until bregma and lambda landmarks were aligned, and then drilled a hole in the skull above the CeA via an electric drill (0.5-mm drill bit, Hartmetall instrumente, HM1005) until the brain tissue was visible. Finally, the thinned skull and residue were removed with fine forceps.

**Viral injections**. A glass capillary (A-M Systems, 626000) was pulled into a micropipette with a neck length of 8–9 mm and a tip diameter of 10 μm. Then a microsyringe (Hamilton, 701 N) was attached to the micropipette and sealed with liquid paraffin, and the microsyringe was loaded via a microsyringe pump. The micropipette tip was immersed in a viral solution consisting of a 1:1 mixture of 200 nl of rAAV-EF1α-DIO-His-EGFP-2a-TVA-WPRE-pA (AAV2/9, 2.00E + 12 vg/mL, BrainVTA PT-0207) and rAAV-EF1α-DIO-RG-WPRE-pA (AAV2/9, 2.00E + 12 vg/mL, BrainVTA PT-0023) (Fig. 1a), and the solution was taken up at a rate of 100 nl/min via a microsyringe pump. The microsyringe was then used to inject the viral solution into the CeA, according to the injection coordinates provided by the mouse brain atlas (vertical injection, AP: −1.0, ML: −2.8, DV: −4.0), at a rate of 20 nl/min. After the injection, the microsyringe remained in place for 10 min and was then slowly pulled out. Finally, the skull surface was cleaned and the incision was sutured. At 3 weeks after the first injection (Fig. 1b), the same procedure was applied to administer another injection of 200 nl of RV-ENVA-ΔG-DsRed (2.00 × 108 infectious unit/mL, BrainVTA R01001) (Fig. 1a).

**Histology**. One week after the final injection of the rabies virus, each mouse was anesthetized by intraperitoneal injection with 2% sodium pentobarbital at a dose of 40 mg/kg, followed by euthanasia via cardiac perfusion. The blood was first replaced with phosphate-buffered saline (PBS), followed by perfusion with 4% paraformaldehyde for tissue fixation. Subsequently, the entire brain was harvested and placed in 4% paraformaldehyde in a refrigerator at 4 °C for 48 h. Thereafter, the post-fixed brain was dehydrated in a 15% sucrose solution until it sank, after which it was immersed in a 30% sucrose solution. After sinking to the bottom again, the brain tissue was placed in a slicing mold and embedded with OCT compound at −20 °C. After solidification, the whole brain was sliced at a thickness of 50 μm by a cryostat (Lecia CM1950). Then the slices were placed in a 48-well plate filled with anti-freezing solution, which was stored in a −20 °C refrigerator for several months.

**CLARITY technique for tissue clearing**. Each whole mouse brain was embedded in 3% agarose and sliced by a vibratome with a thickness of 300 μm. Then, the slices were washed four times in PBS (1 h per wash). After washing, the brain slices were transferred into a 4-ml EP tube with 3 ml of HMS and 1 ml of PBS, and the tube was placed in a refrigerator at 4 °C for 48 h. Thereafter, to polymerize the slices, the EP tube was incubated in a 37 °C water bath for 4 h. After polymerization, the gel wrapped on the surface of the slices was wiped off with a fiber-free tissue. Next, the slices were transferred to a 50-ml EP tube with 30 ml of sodium dodecyl sulfate clean buffer, which was then placed on a shaker at 80–90 rpm for 3d at 37 °C until all the slices were transparent. Finally, the slices were washed three times in PBS (30 min per wash) and were then imaged after immersion in RIMS for 0.5 h.

**Fluorescent in situ hybridization**. Antisense riboprobes (Supplementary Table 4) were reverse transcribed from their target DNA fragments, which were produced from mouse brain cDNA and tagged with the T7 promoter before the 5′-terminal. The riboprobes were labeled with biotin by the T7 reverse transcriptase. All the samples and reagents were prepared and stored in an RNase-free environment. The slices were washed three times in DEPC-PBS (10 min per wash). Then, the slices were treated with 3% hydrogen peroxide for 10 min to block endogenous peroxidases. Next, 0.3% Triton X-100 was also applied for 10 min to permeabilize the tissue. Subsequently, the slices were transferred to acetylation solution, washed three times in DEPC-PBS, and incubated in prehybridization buffer for 1 h in a 60 °C water bath. The target riboprobes were then diluted in the hybridization solution at a concentration of 1 μg/ml and were incubated with the slices for 20 h in a 60 °C water bath. Thereafter, the slices were rinsed three times in 2× SSC (20 min per wash). RNase A was applied at a concentration of 10 μg/ml to digest the excess riboprobes. Then, the slices were rinsed three times in 0.2× SSC for 30 min each time and were then washed three times in PBST (10 min per wash). Next, the slices were incubated with 10% NSS for 1 h, and anti-biotin-pod (1:500) was then added after blocking for overnight incubation at 4 °C. The next morning, the slices were washed three times in PBST (10 min per wash), after which the TSA-Plus Fluorescein system (NEL741B001KT, 1:100, PerkinElmer) was used to detect the primary antibody.

**Immunofluorescent staining**. Sections were washed three times in PBS (10 min per wash), and then they were incubated in 0.3% Triton X-100 to permeabilize for 30 min at 37 °C. Subsequently, 5% donkey serum was employed to block non-specific sites for 30 min at 37 °C. Thereafter, the sections were incubated with primary antibodies for 12–24 h at 4 °C, washed three times in PBS (10 min per wash), and incubated with a corresponding secondary antibody for 2 h at room temperature (Supplementary Table 6 for details). Afterward, the sections were washed three times in PBS (10 min per wash). To determine the brain regions in each slice, fluorescent Nissl counterstaining was applied by NeuroTrace (1:200) for 30 min at room temperature. Finally, all slices were mounted on slides and sealed with 80% glycerin for imaging.

**Image acquisition**. Three imaging strategies were implemented depending on imaging requirements. For imaging of neuronal distributions in each brain region, the Tissue Faxs Plus system (Tissue Genostics, Vienna, Austria) was used to scan and stitch the entire brain slice with a ×10 objective used with a fluorescent microscope. For detailed imaging and morphological analysis of a single neuron, such as that for starter neurons, a laser-scanning confocal system (Zeiss 880, Oberkochen, Germany) was used to collect Z-stack information for 3D-reconstruction. In addition, light-sheet microscopy (VISoR)[58] was also employed to acquire large-scale images without compromising the detail in visualizing dendritic arborizations; this method was specifically used to reconstruct and analyze the morphologies and connectivities of input neurons across brain regions. Two reporter fluorescent proteins, EGFP and DsRed, were excited by 488-nm and 543-nm lasers, respectively, and the wavelength ranges of the emitted fluorescent receivers were 500–550 nm and 570–620 nm, respectively.

**Cell counting**. The starter neurons were defined as the neurons expressing both EGFP and DsRed, and input neurons were defined as the neurons expressing only DsRed but no EGFP. First, we analyzed the distribution of starter neurons. To this aim, we referenced the mouse brain atlas from the Allen Brain Atlas[59,60] (© 2011 Allen Institute for Brain Science. Allen Mouse Brain Atlas. Available from: atlas.brain-map.org/atlas) and The Mouse Brain in Stereotaxic Coordinates[61]. Since the CeA is a relatively small nucleus, and as long as 70% of the starter neurons were located in the CeA, the sample was used for further analysis. Thereafter, we identified the borders of brain regions by fluorescent Nissl counterstaining and divided the brain regions where the input neurons were located according to the brain map provided by the Allen Brain Atlas, after which the input neurons on every other slice were counted. The counting in each brain region was performed automatically using an unbiased stereology TissueFAX Plus ST (Tissue Genostics, Vienna, Austria)[62].

**Morphological features**. The input neurons imaged by our confocal system were reconstructed in Amira 6 (Thermo Fisher Scientific, MA, USA), and their raw data were exported in SWC format, which could then be analyzed in L-measure[63]. The following morphological parameters were directly extracted by L-measure: soma surface, number of stems, number of bifurcations, number of branches, number of tips, width, height, depth, depth-width ratio, Euclidean distance, path distance, and branch path length. Detailed descriptions of these parameters are provided in Supplementary Table 5. Additionally, the somatic depth was measured on the original images. The input neurons imaged by light-sheet microscopy were demonstrated in Imaris (ver. 9.3.1, Bitplane, AG) and reconstructed by Lychins, the latter of which is neuron-tracing software specifically designed for VISoR-imaged neurons.

**Axon pathway reconstruction**. To precisely reconstruct axonal pathways, a digitally operable 3D mouse brain model and its 150 equidistant coronal sections were developed according to the mouse brain atlas of the Allen Brain Institution[60] (© 2011 Allen Institute for Brain Science. Allen Mouse Brain Connectivity Atlas. Available from: connectivity.brain-map.org/3d-viewer). The positions of entire neurons in 3D space were determined by three endpoints of their dendrites and/or axons, which were manually anchored in the coronal sections. By restoring the positions of these three points determined in coronal sections, the entire reconstructed neuron could be displayed in the 3D brain atlas. After loading the brain structures module, the brain nuclei that the fibers passed through and specific pathway information could be comprehensively displayed in 3D (3DS Max, Autodesk, CA, USA). In addition, the coordinated 3D mouse brain model and the platform developed in the present study are available upon request for researchers wanting to track the specific brain regions through which axons pass.

**Statistical and reproducibility**. Correlation analysis and data clusters were generated in MATLAB 9.7 (Mathworks, MA, USA). Pairwise distance between pairs was calculated by the function "pdist", agglomerative hierarchical cluster trees were generated by the function "linkage", then the clusters were constructed from linkages by the function "clusters", dendrograms were plotted by function "dendrogram". All functions are in "Statistics and Machine Learning Toolbox" of MATLAB. Difference calculations were performed by SPSS Statistics 22.0 (IBM, NY, USA). Differences among the three groups were analyzed by one-way analysis of variance, followed by Tukey's post hoc tests. Differences between the two groups were analyzed by Student's $t$ tests. All values are presented as the mean ± standard error of the mean (s.e.m.), and $p$ values <0.05 were considered to represent significant differences between/among groups. All figures were plotted in Prism 8 (GraphPad Software, CA, USA).

**Reporting summary**. Further information on research design is available in the Nature Research Reporting Summary linked to this article.

## Data availability
Source data that support the findings of this study are available in Supplementary Data 1. Any other data are available from the corresponding author on reasonable request.

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

## Acknowledgements
We would like to acknowledge the Mesoscopic Brain Mapping Platform and Open Platform for Brain-inspired Intelligence of NEL-BITA for providing whole-brain VISoR imaging and computation resources, and Qi-Zhi Xu and Jing Wang for their help in developing the coordinated 3D mouse brain model. This work was supported by the National Natural Science Foundation of China (Grant no. 32030046), the Strategic Priority Research Program of the Chinese Academy of Science (Grant no. XDB32020200), the National Key R&D Program of China (Grant no. 2016YFC1305900).

## Author contributions
Conceptualization, C.H. and J.N.Z.; methodology, C.H., Y.W., P.C., and Q.H.S.; software, H.W., and L.F.D.; formal analysis, C.H.; investigation, C.H.; resources, C.H., Y.W., P.C., and Q.H.S.; writing—original draft, C.H.; writing—review & editing, C.H. and J.N.Z.; supervision, J.N.Z., and G.Q.B.; funding acquisition, J.N.Z.

## Competing interests
The authors declare no competing interests.
