## [Peer Review File · Communications Biology]

Reviewers' comments:

Reviewer #1 (Remarks to the Author):

In this study by Huang and Zhou and colleagues, the authors perform a comprehensive analysis of the inputs to central amygdala (CeA) corticotrophin releasing factor-expressing neurons (CRF). For the most part, the extensive analysis is well designed and executed and the results will provide important new insights into the potential function of these cells in a variety of behaviors. Although I am generally supportive of publication, the introduction should be expanded to include the discussion of other papers that have characterized the cell specific function of these neurons, as well as their inputs (see specific comment 1 below). Additionally, there are three conclusions that are made by the authors that are not supported by the data presented. The authors will either need to perform additional experiments to confirm their hypothesis, or not make the specific conclusions that have been drawn.

Specific comments:

1. It is somewhat surprising that the authors state that there is very little know about the function of CeA-CRF neurons, except for a few studies "related to anxiety and conditional fear" and reference three recent papers. There are actually more than 3 studies, 6 that I am aware of, that have investigated the cell-type specific role of CeA-CRF neurons using *Crf-Cre* mice or rats in a wide variety of behaviors including in the modulation of stress, pain, feeding, and alcohol addiction. There is also an even larger body of research on the role of central amygdala CRF with hundreds of publications. The authors should at least reference these cell-type specific papers, including a paper with cell-specific rabies tracing analysis of CeA-CRF neurons that has been previously published (Fadok et al., *Nature*, 2017). Though I will note that this previous paper is not as comprehensive as the study presented here and does not detract from the novelty of this study, the authors should acknowledge this previous work. Some acknowledgement of the decades of research into CeA-CRF should be included, such as through the referencing of some reviews of the subject (e.g. Koob GF. *Brain Res.* 2009 Oct 13; 1293:61-75; Neugebauer V, Mazzitelli M, Cragg B, Ji G, Navratilova E, Porreca F. *Neuropharmacology.* 2020 Jun 15; 170:108052; Gafford GM, Ressler KJ. *Horm Behav.* 2015 Nov; 76:136-42).
2. In figure 3, the authors define convergent versus diffuse inputs to the CeA-CRF neurons. Although it is likely that the convergence model is correct if the input strength is greater than 1, i.e. there are more input neurons than starter neurons, the assumption of diffuse inputs is less straight forward. The authors assume that if the input strength is less than one, then one input neuron must innervate multiple CeA-CRF neurons. However, the methodology employed cannot exclude the possibility that only a subset of CeA-CRF neurons are innervated by the designated input. For example, if the input strength is 0.5, it is possible that on 10% of CeA-CRF neurons are innervated by this input, which would mean that it is also a convergence input.
3. The designation of *CamKIIa* as a largely exclusive marker of glutamatergic neurons is not accurate. *CaMKIIa* is a marker of some, but not all projection neurons that can be either glutamatergic or GABAergic. Because the cortical structures have little to no GABAergic projections, almost all projection neurons that are *CaMKIIa*-positive are glutamatergic, but this does not hold true for projection neurons from subcortical structures. For example, the caudate putamen and the CeA are essentially exclusively GABAergic populations, with a small percentage of cholinergic neurons in the CP. Thus, the *CaMKIIa* neurons that are designated in these regions as glutamatergic would mean that these two structures are ~50% glutamatergic based on the graph in Figure 4 panel B. This is not correct. *Vglut1* and *Vglut2* are not expressed in the CP or CeA and *Vglut3* is only expressed in cholinergic interneurons of the CP (Gras et al., *J. Neurosci.* July 1, 2002, 22(13):5442–5451; see also the Allen Brain Atlas). The cells that did not label with *GAD1* in these regions are likely *GAD2* positive. *Vgat* is a better maker to capture all GABAergic populations. The authors need to modify their conclusions because the results do not fully support their interpretation. Unfortunately, because of the design of the experiment the authors can say very little about the type of projection inputs to the CeA-CRF neurons and all analysis in Figure 4 and Figure S2 are relatively meaningless. To make these conclusions the authors would have to probe for *Vglut1*, *Vglut2*, *Vglut3*, and *Vgat*, Obviously, this experiment is expensive and time consuming. If the authors want to present the data with the many caveats outlined here, and show the data as supplemental material, that would be acceptable, otherwise they will have to redo the experiment with the different marker to directly support their conclusion.
4. The data mapping the trajectory of axons from the SS, MD, and PAG is heroic and beautiful.

However, the juxtaposition of the axon to PSD-95 presented in Figure 7 is not sufficient to conclude that these cells form functional en passant synapses. The authors would have to express ChR2 in neurons projecting to the CeA and patch in these regions of passage and demonstrate the presence of light-evoked EPSCs or IPSCs. As it stands, this data could be presented in supplemental material and the authors could suggest that these cells may form en passant synapses, but they cannot conclude that they do.

Reviewer #2 (Remarks to the Author):

This study identified and characterized brain-wide monosynaptic inputs to CeA CRF neurons. This was achieved using Cre-dependent rabies tracing in CRF-Cre mice. A number of fascinating patterns and properties of input regions and cell-types were discovered. Creative analyses were performed that provide key clues as to how CeA CRF neurons may be engaged to regulate a range of behavioral outputs, as well insight into how groups of upstream regions may modulate them in concert. This manuscript provides a large catalogue of data that will likely inform new studies on CRF neurons and help generate novel hypotheses. While overall the findings and structure of the paper are appropriate, there are several concerns related to study rigor and data interpretation.

The CeA, and the CRF neurons within, have both anterior and posterior aspects, for which some evidence exists that there may be different inputs and outputs. The injection site reported here is AP -1.0, yet the CeA extends to almost AP -2.0. Authors should provide the extent to which helper viruses infected CeA CRF neurons in each subject for proper interpretation of the inputs. Are the starter neuron distributions across the AP axis uniform across mice?

Lines 312-314 of the discussion mention "control experiments", but these are nowhere to be found in the manuscript. Data showing lack of infection of CRF- neurons by EnvA and lack of spread of EnvA without dG should be included in the supplement.

There is no information in the methods on specific clustering analyses or algorithms used to identify the 6 clusters of input regions. "The 44 subregions were clustered into six groups according to spearman correlations and hierarchal clustering (average method)" is not sufficient. Analysis packages, plugins, or functions used in MATLAB should to be reported.

Data presented in Figure 3 are fascinating and important for hypothesizing how different streams of information are delivered to CeA. However, the distribution in Figure 3C may prove specious when considering the CeA contribution to the STR group. It is possible that many of these "upstream" counts are just starter neurons or locally infected dsRed interneurons. To this end, there is no information on whether cell counting specifically in the CeA excludes EGFP+ starter cells. Redoing this analysis while excluding the CeA neurons will provide a cleaner map of long-range input distributions and strengths.

Regarding Figure 4, CaMKII is a reliable marker for excitatory neurons in cortex, yet this rule breaks down subcortically where isoforms of CaMKII are not exclusive to excitatory neurons. Authors should reanalyze data from all non-cortical structures here and use either a more reliable excitatory marker (Vglut2) or GAD- cells as an index. The striatum and CeA are good examples because they contain GABAergic cells that express CaMKII. At the minimum it will be important to show that dsRed input cells that are CaMKII+ are indeed GAD- in the CP and CeA.

Authors should cite Fadok et al., Nature 2017 and acknowledge that the limited analysis in this paper is consistent with the current dataset.

No speculation as to what the "others" neuron subtypes might be.

Reviewer #3 (Remarks to the Author):

Huang and colleagues focus on the dense population of CRF-expressing neurons in the CeA to construct a whole brain map of monosynaptic inputs to this specific subpopulation of CeA neurons. To this end, the authors used rabies virus-assisted retrograde tracing in combination with CRF-Cre mice. Their thorough analysis identified 44 subregions that provide input to CeA-CRF neurons. The authors carefully quantified the input strengths taking into account the number of starter neurons and discriminated between excitatory and inhibitory inputs. Finally, they characterized the morphological features of those input cells and reconstructed the long-range projections of individual neurons. This is a comprehensive and thorough study of high technical quality. This data set can serve as an important resource to further disentangle the specific role of CeA-CRF neurons in fear and anxiety.

I have the following suggestions and recommendations:

1. The correct designation of CRF is "corticotropin-releasing factor" and not "corticotrophin-releasing factor".
2. The authors mention that they performed several control experiments to ensure the quality and reliability of their rabies virus-based retrograde tracing, e.g., by ruling out that the rabies virus was not able to infect CRF negative neurons. Another informative control would have been the conformation of some of those distant input regions by anterograde tracing. At least for excitatory neurons, this would be an easy endeavor as one could use rAAVs, e.g., expressing synaptophysin-GFP or -RFP driven by a CaMKII promoter.
3. The utilization of CaMKII as a marker of excitatory neurons has some flaws as there are populations of GABAergic projection neurons which also express CaMKII such as medium spiny neurons in the caudate putamen. Other markers like Vglut1-3 would have provided more specificity in this respect. How can the authors rule out that in particular the CaMKII neurons found in the STR-related nuclei are not also GABAergic?
4. Along these lines, the authors state that the GAD-labeled GABAergic neurons in the cortex were all long-range projecting neurons. What about the other GABAergic neurons found at distant sites? Are these also long-range projecting neurons or can they still be defined as proper interneurons? This is an aspect the authors have to discuss in more detail.
5. Depending on the input strength, the authors define two groups of neurons, i.e., the convergent and the diffuse group. To my understanding, this extrapolation is only valid under the assumption that all starter neurons receive more or less equal input. However, this assumption would be violated if there were major differences in the afferents CeA-CRF neurons receive. In this respect, the authors might want to discuss potential differences between local interneurons and projection neurons existing among CeA-CRF neurons.
6. One aspect the authors entirely neglect in their discussion is a direct comparison of CeA-CRF inputs with those of other populations of CeA neurons, e.g., SST1- or PKCdelta-expressing neurons, e.g., compare Fu et al., 2020 Whole-Brain Map of Long-Range Monosynaptic Inputs to Different Cell Types in the Amygdala of the Mouse. *Neurosci Bull.* 36(11):1381-1394.

Minors:

1. As the authors use only a Gad1-specific riboprobe, I would recommend to specify this throughout the manuscript instead of using the more general term "GAD".
2. Page 1, line 14: It seems that the 4th affiliation is not correct: Institute of Artificial Institute.
3. Page 4, line 62: Neuropeptides such as CRF are rather considered as a neuromodulators than as neurotransmitters. Therefore, substitute "neurotransmitter" by "neuromodulator".
4. Page 41, line 810: Substitute "envelops protein" by "envelope protein".

Reviewer comments:

Reviewer #1:

We thank the reviewer for calling our work “well-designed and executed”. We appreciate the statement that “the results will provide important new insights into the potential function of these cells in a variety of behaviors” and hope our responses to the queries below are satisfactorily.

Remark #1:

It is somewhat surprising that the authors state that there is very little know about the function of CeA-CRF neurons, except for a few studies “related to anxiety and conditional fear” and reference three recent papers. There are actually more than 3 studies, 6 that I am aware of, that have investigated the cell-type specific role of CeA-CRF neurons using Crf-Cre mice or rats in a wide variety of behaviors including in the modulation of stress, pain, feeding, and alcohol addiction. There is also an even larger body of research on the role of central amygdala CRF with hundreds of publications. The authors should at least reference these cell-type specific papers, including a paper with cell-specific rabies tracing analysis of CeA-CRF neurons that has been previously published (Fadok et al., Nature, 2017). Though I will note that this previous paper is not as comprehensive as the study presented here and does not detract from the novelty of this study, the authors should acknowledge this previous work. Some acknowledgement of the decades of research into CeA-CRF should be included, such as through the referencing of some reviews of the subject (e.g. Koob GF. Brain Res. 2009 Oct 13;1293:61-75; Neugebauer V, Mazzitelli M, Cragg B, Ji G, Navratilova E, Porreca F. Neuropharmacology. 2020 Jun 15;170:108052; Gafford GM, Ressler KJ. Horm Behav. 2015 Nov;76:136-42).

Reply to #1:

Thank you for this constructive suggestion.

We have added the references that reviewer mentioned, which is described below and now also included in the introduction section (Page 3-4, Line 56-67), as follows:

In addition to neuronal subtypes expressing either somatostatin or protein kinase C- δ in inhibitory circuits encoding fear (Penzo et al., 2015), peptide-expressing neurons in the CeA have recently been investigated (Neugebauer et al., 2020). Corticotrophin-releasing factor (CRF) is a stress-related peptide that is expressed in a large subpopulation of CeA neurons (Wang et al., 2021). Interestingly, CeA-CRF neurons represent one of the most densely distributed populations of CRF neurons throughout the brain (Peng et al., 2017), which has attracted the attention of many

research groups (Gafford and Ressler, 2015; Koob, 2009). By employing CRF-Cre mice or rats, it has been discovered that they are involved in mediating stress (McCall et al., 2015), pain (Andreoli et al., 2017), alcohol addiction (de Guglielmo et al., 2019) and fear (Jo et al., 2020; Sanford et al., 2017). In addition, by injecting the RV virus, a fraction of putative excitatory input brain regions was retrogradely traced in a previous research, which is consistent with the current dataset (Fadok et al., 2017).

Remark #2:

In figure 3, the authors define convergent versus diffuse inputs to the CeA-CRF neurons. Although it is likely that the convergence model is correct if the input strength is greater than 1, i.e. there are more input neurons than starter neurons, the assumption of diffuse inputs is less straight forward. The authors assume that if the input strength is less than one, then one input neuron must innervate multiple CeA-CRF neurons. However, the methodology employed cannot exclude the possibility that only a subset of CeA-CRF neurons are innervated by the designated input. For example, if the input strength is 0.5, it is possible that on 10% of CeA-CRF neurons are innervated by this input, which would mean that it is also a convergence input.

Reply to #2:

It's a very incise comment on our models of input patterns. As the example that reviewer described, we assume that there are 100 input neurons in region A, and 200 starters in this CeA, these inputs only project to 10% of the starters (20 starters), which results in a convergent model. However, in this case, these 20 starters are the true starters for these 100 input neurons, and the input strength of region A should be corrected to 5, but not 0.5. Therefore, the only problem here is that we cannot distinguish the true starter from the false positive starters for each region by current methodology, and by default all starters accept the projections from the input neurons. Nevertheless, we have elaborated this possibility and the flaw of our theory in the discussion (Page 18, Line 352-358), which is also described below:

In the convergent pattern, more than one input neuron in a given subregion innervated a single starter neuron, and the majority of brain regions with this input pattern innervated starter neurons in a top-down manner. On the contrary, in the diffuse pattern, a single input neuron could probably project to multiple starter neurons. However, due to the limitations of the methodology in this study, we cannot examine every single starter that the inputs innervate. More specifically, there may be some of the inputs in the diffuse group that only project to a fraction of the starters, whose number is even smaller than the inputs, resulting in a convergent pattern.

Remark #3:

The designation of CamKIIa as a largely exclusive marker of glutamatergic neurons is not accurate. CaMKIIa is a marker of some, but not all projection neurons that can be either glutamatergic or GABAergic. Because the cortical structures have little to no GABAergic projections, almost all projection neurons that are CaMKIIa-positive are glutamatergic, but this does not hold true for projection neurons from subcortical structures. For example, the caudate putamen and the CeA are essentially exclusively GABAergic populations, with a small percentage of cholinergic neurons in the CP. Thus, the CaMKIIa neurons that are designated in these regions as glutamatergic would mean that these two structures are ~50% glutamatergic based on the graph in Figure 4 panel B. This is not correct. Vglut1 and Vglut2 are not expressed in the CP or CeA and Vglut3 is only expressed in cholinergic interneurons of the CP (Gras et al., J. Neurosci. July 1, 2002, 22(13):5442–5451; see also the Allen Brain Atlas). The cells that did not label with GAD1 in these regions are likely GAD2 positive. Vgat is a better maker to capture all GABAergic populations. The authors need to modify their conclusions because the results do not fully support their interpretation. Unfortunately, because of the design of the experiment the authors can say very little about the type of projection inputs to the CeA-CRF neurons and all analysis in Figure 4 and Figure S2 are relatively meaningless. To make these conclusions the authors would have to probe for Vglut1, Vglut2, Vglut3, and Vgat, Obviously, this experiment is expensive and time consuming. If the authors want to present the data with the many caveats outlined here, and show the data as supplemental material, that would be acceptable, otherwise they will have to redo the experiment with the different marker to directly support their conclusion.

Reply to #3:

The reviewer has raised a fabulous question on our neuron-type characterization results. Indeed, there is no single marker that can characterize all excitatory neurons or inhibitory neurons in the central nervous system. To demonstrate whether

subcortical CaMKII-expressing input neuron might be inhibitory neurons, we have carried out a new experiment that we directly label GABA by immunofluorescence and find that some CaMKII-expressing input neuron colocalized with GABA in CP and CeA (Fig. S4M, shown below). Therefore, in order to be more accurate, we have replaced the expression of “excitatory neurons” with “CaMKII-expressing neuron”, and the expression of “inhibitory neurons” are replaced by the “GAD1-expressing neurons” in the results section. In addition, we have discussed these caveats in the discussion section as suggested by the reviewers (Page 19, Line 372-381), which is also described below:

In our present study, CaMKII was used as a marker for excitatory neurons in the cortical regions, while inhibitory neurons may be identified by GAD1 expression. However, there is no single marker that can represent either all excitatory neurons or inhibitory neurons (Rees et al., 2017). That said, CaMKII-positive neurons account for over 70% of hippocampal pyramidal neurons and colocalize with most glutamatergic neurons in the neocortex and other brain regions. Notably, CaMKII is not a reliable marker of excitatory neurons in subcortical regions (Fig. S4M). Meanwhile, GAD1-positive neurons mainly represent inhibitory interneurons (Wang et al., 2013). Collectively, these known features account for CaMKII-positive neurons and GAD1-positive neurons in our present study not fully recapitulating all our labeled input neurons, therefore, those other neurons may also be excitatory neurons that express Vglut but not CaMKII (Gras et al., 2002), and the CaMKII-expressing neurons in subcortical regions may be inhibitory neurons.

Fig. S4M: The input neurons in CP (left) and CeA (right) were labelled by both CaMKII FISH probe and anti-GABA antibody, and several CaMKII-expressing inputs were colocalized with the GABA immunofluorescent both in CP and CeA, indicating that the CaMKII is not an accurate marker of excitatory neurons in subcortical regions.

Remark #4:

4. The data mapping the trajectory of axons from the SS, MD, and PAG is heroic and beautiful. However, the juxtaposition of the axon to PSD-95 presented in Figure 7 is not sufficient to conclude that these cells form functional en passant synapses. The authors would have to express ChR2 in neurons projecting to the CeA and patch in these regions of passage and demonstrate the presence of light-evoked EPSCs or IPSCs. As it stands, this data could be presented in supplemental material and the authors could suggest that these cells may form en passant synapses, but they cannot conclude that they do.

Reply to #4:

Thank you for commenting that our data mapping the trajectory of axons is heroic and beautiful.

We hypothesize that these *en passant* fibers are rather innervated by the neurons than projected to the neurons in the passing regions, and the *en passant* fibers not encapsulated by MBP as well as the postsynaptic marker of PSD95 on these fibers indicate this possibility. But due to the limitation of current methodology, there is no way that can express ChR2 in the neurons that specifically project fiber to innervate the *en passant* axons. Therefore, we speculate that these *en passant* fibers may form synapses and this part of result has been moved to the supplementary material in Fig. S8.

Reviewer #2:

We thank the reviewer for commenting our “fascinating patterns and properties”, “Creative analyses” and calling our manuscript “provides a large catalogue of data that will likely inform new studies on CRF neurons and help generate novel hypotheses”.

Remark #1:

The CeA, and the CRF neurons within, have both anterior and posterior aspects, for which some evidence exists that there may be different inputs and outputs. The injection site reported here is AP -1.0, yet the CeA extends to almost AP -2.0. Authors should provide the extent to which helper viruses infected CeA CRF neurons in each subject for proper interpretation of the inputs. Are the starter neuron distributions across the AP axis uniform across mice?

Reply to #1:

The reviewer has raised a very important question.

To answer this question, we have carried out a new experiment to examine the distribution of the starters in CeA and the representative images are shown below, which is now included in Fig. S2A. The numbers of starters in each subject are counted from anterior to posterior and shown in Fig. S2B. The starter neurons are centrally distributed around the injection site, and the distributions across the AP axis are uniform across subjects.

Fig. S2. Most starter neurons were located in the CeA.

(A) To verify the reliability of the approach in every tracing case, the whole brain of each CRF-Cre mouse in the virus-tracing group was sectioned to examine the morphological distribution of starter neurons in the amygdala. Representative distribution of starter neurons in 50- μ m serial-scanned images. The position of bregma is shown in the upper-right corner to indicate the interval between each slice.

(B) The distribution of starter neurons, from anterior to posterior, within the CeA. Most of the starter neurons ($67.87 \pm 3.31\%$) were in the posterior and middle parts of the CeA.

Remark #2:

Lines 312-314 of the discussion mention “control experiments”, but these are nowhere to be found in the manuscript. Data showing lack of infection of CRF-neurons by EnvA and lack of spread of EnvA without dG should be included in the

supplement.

Reply to #2:

We are sorry for our carelessness about missing the results of “control experiments” in the last version. As the reviewer suggested, the control experiments have been demonstrated below, which is now included in Fig. S1. Fig. S1A shows that in the absence of RVG with only TVA, RV infection is limited to neurons near the injection site that expressed TVA. On the contrary, Fig. S1B shows that in the absence of TVA with only RVG, RV is unable to infect any neuron.

Fig. S1. Representative images of experimental controls.

(A) To confirm the necessity of RG to RV transsynaptic transmission, only one of the two helper viruses (AAV-DIO-TVA-EGFP, green) and helper-dependent RV (red) were injected into the CeA of CRF-Cre mice, which resulted in neurons that coexpressed both EGFP and DsRed without extrinsic input only expressed as DsRed. This result indicated that RG was a key component for RV to retrogradely infect from starter neurons.

(B) Similarly, only injecting one of the two helper viruses (AAV-DIO-RG) and helper-dependent RV did not result in any DsRed-labeled neurons, indicating that TVA was necessary for RV to infect starter neurons.

Remark #3:

There is no information in the methods on specific clustering analyses or algorithms used to identify the 6 clusters of input regions. “The 44 subregions were clustered into six groups according to spearman correlations and hierarchal clustering (average method)” is not sufficient. Analysis packages, plugins, or functions used in MATLAB should be reported.

Reply to #3:

We thank the reviewer for pointing this out and the specific method on clustering analyses is added in method section, statistical analysis part (Page 28, Line 568-572) as follows:

Correlation analysis and data clusters were generated in MATLAB 9.7 (Mathworks, MA, USA). Pairwise distance between pairs were calculated by the function “pdist”, agglomerative hierarchical cluster trees were generated by the function “linkage”, then the clusters were constructed from linkages by the function “clusters”, dendrograms were plotted by function “dendrogram”. All functions are in “Statistics and Machine Learning Toolbox” of MATLAB.

Remark #4:

Data presented in Figure 3 are fascinating and important for hypothesizing how different streams of information are delivered to CeA. However, the distribution in Figure 3C may prove specious when considering the CeA contribution to the STR group. It is possible that many of these “upstream” counts are just starter neurons or locally infected dsRed interneurons. To this end, there is no information on whether cell counting specifically in the CeA excludes EGFP+ starter cells. Redoing this analysis while excluding the CeA neurons will provide a cleaner map of long-range input distributions and strengths.

Reply to #4:

Thanks to the reviewer for commenting that our hypothesis is fascinating and important.

We apologize for this unclear description on the strategy of the cell counting of starter and input neurons in the last version. In fact, the analysis was performed as reviewer suggested, i.e., the starter neurons were identified by the coexpression of EGFP and DsRed around the injection site, and the input neurons that directly project to CeA-CRF neurons were identified by the expression of only DsRed but no EGFP (Sun et al., 2019; Watabe-Uchida et al., 2012). We have provided more information on the cell counting in results (Page 6, Line 108-110) and method (Page 27, Line 534-535) sections.

Remark #5:

Regarding Figure 4, CaMKII is a reliable marker for excitatory neurons in cortex, yet this rule breaks down subcortically where isoforms of CaMKII are not exclusive to excitatory neurons. Authors should reanalyze data from all non-cortical structures here and use either a more reliable excitatory marker (Vglut2) or GAD- cells as an

index. The striatum and CeA are good examples because they contain GABAergic cells that express CaMKII. At the minimum it will be important to show that dsRed input cells that are CaMKII+ are indeed GAD- in the CP and CeA.

Reply to #5:

Thanks to the reviewer for this incisive question. To answer this question, we have carried out a new experiment that directly labeled the CaMKII-expressing input neurons by the anti-GABA antibody in the CP and CeA. However, some of the CaMKII-expressing inputs in CP and CeA are colocalized with the immunofluorescence of GABA (Fig. S4M), which is shown below. Therefore, we have changed the expression of “excitatory / inhibitory neurons” into “CaMKII-expressing / GAD1-expressing neurons” in results section and discussed these break-down rules in discussion (Page 19, line 372-381), which is shown as follows:

In our present study, CaMKII was used as a marker for excitatory neurons in the cortical regions, while inhibitory neurons may be identified by GAD1 expression. However, there is no single marker that can represent either all excitatory neurons or inhibitory neurons (Rees et al., 2017). That said, CaMKII-positive neurons account for over 70% of hippocampal pyramidal neurons and colocalize with most glutamatergic neurons in the neocortex and other brain regions. Notably, CaMKII is not a reliable marker of excitatory neurons in subcortical regions (Fig. S4M). Meanwhile, GAD1-positive neurons mainly represent inhibitory interneurons (Wang et al., 2013). Collectively, these known features account for CaMKII-positive neurons and GAD1-positive neurons in our present study not fully recapitulating all our labeled input neurons, therefore, those other neurons may also be excitatory neurons that express Vglut but not CaMKII (Gras et al., 2002), and the CaMKII-expressing neurons in subcortical regions may be inhibitory neurons.

Fig S4M: The input neurons in CP (left) and CeA (right) were labelled by both CaMKII FISH probe and anti-GABA antibody, and several CaMKII-expressing inputs were colocalized with the GABA immunofluorescent both in CP and CeA, indicating

that the CaMKII is not an accurate marker of excitatory neurons in subcortical regions.

Remark #6:

Authors should cite Fadok et al., Nature 2017 and acknowledge that the limited analysis in this paper is consistent with the current dataset.

Reply to #6:

Thanks to the reviewer for reminding of the previous finding by Fadok et al. We have cited this paper in introduction (Page 4, Line 65-67), which is shown below as follow:

In addition, by injecting the RV virus, a fraction of putative excitatory input brain regions was retrogradely traced in a previous research, which is consistent with the current dataset (Fadok et al., 2017).

Remark #7:

No speculation as to what the “others” neuron subtypes might be.

Reply to #7:

It’s a fascinating question that the reviewer asked. We speculate those “other” neurons may be excitatory neurons that express Vglut but not CaMKII. This speculation have been added in discussion (Page 19, Line 378-380), which is described below as follow:

Collectively, these known features account for CaMKII-positive neurons and GAD1-positive neurons in our present study not fully recapitulating all our labeled input neurons, therefore, those other neurons may also be excitatory neurons that express Vglut but not CaMKII.

Reviewer #3:

We thank the reviewer for commenting our study is “comprehensive and thorough with high technical quality” and our data set “can serve as an important resource to further disentangle the specific role of CeA-CRF neurons in fear and anxiety”.

Remark #1:

The correct designation of CRF is “corticotropin-releasing factor” and not “corticotrophin-releasing factor”.

Reply to #1:

Thanks to the reviewer for this correction, all spelling has been checked and corrected.

Remark #2:

The authors mention that they performed several control experiments to ensure the quality and reliability of their rabies virus-based retrograde tracing, e.g., by ruling out that the rabies virus was not able to infect CRF negative neurons. Another informative control would have been the conformation of some of those distant input regions by anterograde tracing. At least for excitatory neurons, this would be an easy endeavor as one could use rAAVs, e.g., expressing synaptophysin-GFP or -RFP driven by a CaMKII promoter.

Reply to #2:

Thanks to the reviewer for this instructive suggestion. We have performed several new control experiments and the results are shown below. In Fig. S1C, it shows that the rabies virus was not able to infect CRF negative neurons, and the representative image in Fig. S1D shows that the starters were CRF positive neurons. As for the anterograde tracing, we have used a virus that reviewer suggested (rAAV-CaMKIIa-EGFP-WPRE-hGH-pA), and have injected in MOs, the projecting fibers are found in CeA. These results are presented in Fig S1E.

Fig. S1 (C) Injecting both of the two helper viruses (AAV-DIO-TVA-EGFP, AAV-DIO-RG) and helper-dependent RV into wild-type mice also did not result in DsRed-labeled neurons, indicating that all helper viruses expressed all components strictly depending on the expression of Cre recombinase.

(D) To further confirm the specificity of the neuronal types of starter neurons, we performed FISH to identify the neuron-subtype of starter neurons. Most of the starter neurons were positive for the CRF mRNA probe, which indicated that the starter neurons were CRF-expressing neurons and that the CRF-Cre line that we used had a high specificity. Left: Representative tile scan image in the CeA. Right: Most of the starter neurons that coexpressed helper viruses (red) and RV (green) were CRF-positive neurons (blue).

(E) An rAAV expressing EGFP driven by CaMKII were injected in MOs, where the CaMKII-expressing inputs mainly distributed. The terminal fibers can be clearly observed in CeA, indicating that there were inputs expressing CaMKII projecting to central amygdala.

Remark #3:

The utilization of CaMKII as a marker of excitatory neurons has some flaws as there are populations of GABAergic projection neurons which also express CaMKII such as medium spiny neurons in the caudate putamen. Other markers like Vglut1-3 would have provided more specificity in this respect. How can the authors rule out that in particular the CaMKII neurons found in the STR-related nuclei are not also GABAergic?

Reply to #3:

It's a very valuable question that reviewer asked. To answer this question, we have carried out a new experiment that directly labeled the CaMKII-expressing input neurons by anti-GABA antibody in Caudoputamen, which is shown below and included in Fig. S4M. However, we have found that several CaMKII-expressing inputs are colocalized with GABA in CP. In addition, there are little Vglut1 and Vglut2 expression in CP, and vglut3 is also expressed in a small amount according to the Allen Brain Atlas, which is shown below. Therefore, to be more accurate, we have changed the expression of “excitatory / inhibitory neurons” into “CaMKII-expressing / GAD1-expressing neurons” in results section and discussed these differences in discussion (Page 19, line 372-381), which is also shown as follows:

In our present study, CaMKII was used as a marker for excitatory neurons in the cortical regions, while inhibitory neurons may be identified by GAD1 expression. However, there is no single marker that can represent either all excitatory neurons or inhibitory neurons (Rees et al., 2017). That said, CaMKII-positive neurons account for over 70% of hippocampal pyramidal neurons and colocalize with most glutamatergic neurons in the neocortex and other brain regions. Notably, CaMKII is not a reliable marker of excitatory neurons in subcortical regions (Fig. S4M). Meanwhile, GAD1-positive neurons mainly represent inhibitory interneurons (Wang et al., 2013). Collectively, these known features account for CaMKII-positive neurons and GAD1-positive neurons in our present study not fully recapitulating all our labeled input neurons, therefore, those other neurons may also be excitatory neurons that express Vglut but not CaMKII (Gras et al., 2002), and the CaMKII-expressing neurons in subcortical regions may be inhibitory neurons.

Fig. S4M: The input neurons in CP (left) and CeA (right) were labelled by both CaMKII FISH probe and anti-GABA antibody, and several CaMKII-expressing inputs were colocalized with the GABA immunofluorescent both in CP and CeA, indicating that the CaMKII is not an accurate marker of excitatory neurons in subcortical regions.

Specimen Age: P56 Vglut1

Symbol: Slc17a7
solute carrier family 17
(sodium-dependent
inorganic phosphate
cotransporter), member
7

Specimen Age: P56 Vglut2

Symbol: Slc17a6
solute carrier family 17
(sodium-dependent
inorganic phosphate
cotransporter), member
6

Specimen Age: P56 Vglut3

Symbol: Slc17a8
solute carrier family 17
(sodium-dependent
inorganic phosphate
cotransporter), member
8

In-situ images of Vglut1-3 in Caudoputamen from Allen Brain Atlas shows that there are little Vglut1 and Vglut2 expression in CP, and vglut3 is also expressed in a small amount.

Remark #4:

Along these lines, the authors state that the GAD-labeled GABAergic neurons in the cortex were all long-range projecting neurons. What about the other GABAergic

neurons found at distant sites? Are these also long-range projecting neurons or can they still be defined as proper interneurons? This is an aspect the authors have to discuss in more detail.

Reply to #4:

The reviewer has asked a thought-provoking question. We believe that the other GABAergic neurons found at distant sites are also long-range projecting neurons, which is also a subset of interneurons in cortex (long-range interneuron). Traditionally speaking, interneurons, whose axons and dendrites are locally distributed to send inhibitory signals, are distinguished from principal neurons in cortex (Ascoli et al., 2008). However, recent studies show that there are some inhibitory neurons whose short-range axon collaterals comply with the traditional definition that innervate local neurons, but they also send long-range axons project to other regions, such as the amygdala (Lee et al., 2014; Tremblay et al., 2016). As suggested by the reviewer, we have added this point in discussion (Page 19, Line 381-386), which is shown as follows:

Nevertheless, we found that GAD1-labeled GABAergic neurons at distance were all long-range-projecting neurons targeting CeA-CRF neurons. This finding is consistent with previous reports showing that GABAergic long-range-projecting neurons in cortex preferentially target inhibitory interneurons in subcortical regions, which allows them to control remote target areas via disinhibition (Melzer and Monyer, 2020), such as amygdala (Lee et al., 2014). And the GABAergic long-range-projection neurons were also found in subcortical regions (Lee et al., 2014).

Remark #5:

Depending on the input strength, the authors define two groups of neurons, i.e., the convergent and the diffuse group. To my understanding, this extrapolation is only valid under the assumption that all starter neurons receive more or less equal input. However, this assumption would be violated if there were major differences in the afferents CeA-CRF neurons receive. In this respect, the authors might want to discuss potential differences between local interneurons and projection neurons existing among CeA-CRF neurons.

Reply to #5:

Thanks to the reviewer for this insightful suggestion. This hypothesis is based on the group of input neurons or starter neurons. Due to the limitation of the methodology in our study, we cannot analyze the input differences between each different starter. Therefore, some input neurons in convergent group may be the local interneurons that innervate many starters, and some input neurons in diffuse group may be the

projection neurons that project to a single starter. We have discussed the possibility that the reviewer mentioned in discussion section (Page 18, Line 352-358) as follows:

In the convergent pattern, more than one input neuron in a given subregion innervated a single starter neuron, and the majority of brain regions with this input pattern innervated starter neurons in a top-down manner. On the contrary, in the diffuse pattern, a single input neuron could probably project to multiple starter neurons. However, due to the limitations of the methodology in this study, we cannot examine every single starter that the inputs innervate. More specifically, there may be some of the inputs in the diffuse group that only project to a fraction of the starters, whose number is even smaller than the inputs, resulting in a convergent pattern.

Remark #6:

One aspect the authors entirely neglect in their discussion is a direct comparison of CeA-CRF inputs with those of other populations of CeA neurons, e.g., SST1- or PKCdelta-expressing neurons, e.g., compare Fu et al., 2020 Whole-Brain Map of Long-Range Monosynaptic Inputs to Different Cell Types in the Amygdala of the Mouse. *Neurosci Bull.* 36(11):1381-1394.

Reply to #6:

The reviewer made a thoughtful suggestion. We have made a careful comparison among these three input atlases in the discussion section as the reviewer suggested (Page 16-17-18, Line 341-347), which is also shown below as follows:

There are multiple neuronal subtypes with different molecular markers distributed in CeA (McCullough et al., 2018). In addition to CRF, a previous study has mapped the monosynaptic input atlas of somatostatin (SST) and protein kinase C- δ (PKC- δ) at whole brain scale (Fu et al., 2020). All three molecularly distinct neurons within CeA receive extensive monosynaptic information throughout the brain, with inputs distributed from the anterior olfactory areas to the posterior midbrain regions. In comparison, CeA-SST neurons receive more inputs from the cortex, CeA-PKC- δ neurons receive more inputs from the striatum, and CeA-CRF neurons receive their main inputs from both cortex and striatum.

Remark Minors:

1. As the authors use only a Gad1-specific riboprobe, I would recommend to specify this throughout the manuscript instead of using the more general term “GAD”.
2. Page 1, line 14: It seems that the 4th affiliation is not correct: Institute of Artificial Institute.

3. Page 4, line 62: Neuropeptides such as CRF are rather considered as a neuromodulators than as neurotransmitters. Therefore, substitute “neurotransmitter” by “neuromodulator”.

4. Page 41, line 810: Substitute “envelops protein” by “envelope protein”.

Reply to Minors:

Thanks to the reviewer for these constructive comments. All these suggestions have been carefully treated and corrected in the manuscript.

Reference

- Andreoli, M., Marketkar, T., and Dimitrov, E. (2017). Contribution of amygdala CRF neurons to chronic pain. *Exp Neurol* 298, 1-12.
- Ascoli, G.A., Alonso-Nanclares, L., Anderson, S.A., Barrionuevo, G., Benavides-Piccione, R., Burkhalter, A., Buzsáki, G., Cauli, B., Defelipe, J., Fairén, A., *et al.* (2008). Petilla terminology: nomenclature of features of GABAergic interneurons of the cerebral cortex. *Nature reviews Neuroscience* 9, 557-568.
- de Guglielmo, G., Kallupi, M., Pomrenze, M.B., Crawford, E., Simpson, S., Schweitzer, P., Koob, G.F., Messing, R.O., and George, O. (2019). Inactivation of a CRF-dependent amygdalofugal pathway reverses addiction-like behaviors in alcohol-dependent rats. *Nat Commun* 10, 1238.
- Fadok, J.P., Krabbe, S., Markovic, M., Courtin, J., Xu, C., Massi, L., Botta, P., Bylund, K., Muller, C., Kovacevic, A., *et al.* (2017). A competitive inhibitory circuit for selection of active and passive fear responses. *Nature* 542, 96-100.
- Fu, J.Y., Yu, X.D., Zhu, Y., Xie, S.Z., Tang, M.Y., Yu, B., and Li, X.M. (2020). Whole-Brain Map of Long-Range Monosynaptic Inputs to Different Cell Types in the Amygdala of the Mouse. *Neuroscience bulletin* 36, 1381-1394.
- Gafford, G.M., and Ressler, K.J. (2015). GABA and NMDA receptors in CRF neurons have opposing effects in fear acquisition and anxiety in central amygdala vs. bed nucleus of the stria terminalis. *Horm Behav* 76, 136-142.
- Gras, C., Herzog, E., Bellenchi, G.C., Bernard, V., Ravassard, P., Pohl, M., Gasnier, B., Giros, B., and El Mestikawy, S. (2002). A third vesicular glutamate transporter expressed by cholinergic and serotonergic neurons. *The Journal of neuroscience : the official journal of the Society for Neuroscience* 22, 5442-5451.
- Jo, Y.S., Namboodiri, V.M.K., Stuber, G.D., and Zweifel, L.S. (2020). Persistent activation of central amygdala CRF neurons helps drive the immediate fear extinction deficit. *Nat Commun* 11, 422.
- Koob, G.F. (2009). Brain stress systems in the amygdala and addiction. *Brain Res* 1293, 61-75.
- Lee, A.T., Vogt, D., Rubenstein, J.L., and Sohal, V.S. (2014). A class of GABAergic neurons in the prefrontal cortex sends long-range projections to the nucleus accumbens and elicits acute avoidance behavior. *The Journal of neuroscience : the official journal of the Society for Neuroscience* 34, 11519-11525.
- McCall, J.G., Al-Hasani, R., Siuda, E.R., Hong, D.Y., Norris, A.J., Ford, C.P., and Bruchas, M.R. (2015). CRH Engagement of the Locus Coeruleus Noradrenergic System Mediates Stress-Induced Anxiety. *Neuron* 87, 605-620.
- McCullough, K.M., Morrison, F.G., Hartmann, J., Carlezon, W.A., Jr., and Ressler, K.J. (2018). Quantified Coexpression Analysis of Central Amygdala Subpopulations. *eNeuro* 5.
- Melzer, S., and Monyer, H. (2020). Diversity and function of corticopetal and corticofugal GABAergic projection neurons. *Nature reviews Neuroscience* 21, 499-515.
- Neugebauer, V., Mazzitelli, M., Cragg, B., Ji, G., Navratilova, E., and Porreca, F. (2020). Amygdala, neuropeptides, and chronic pain-related affective behaviors. *Neuropharmacology* 170, 108052.
- Peng, J., Long, B., Yuan, J., Peng, X., Ni, H., Li, X., Gong, H., Luo, Q., and Li, A. (2017). A Quantitative Analysis of the Distribution of CRH Neurons in Whole Mouse Brain. *Front Neuroanat* 11, 63.
- Penzo, M.A., Robert, V., Tucciarone, J., De Bundel, D., Wang, M., Van Aelst, L., Darvas, M., Parada,

- L.F., Palmiter, R.D., He, M., *et al.* (2015). The paraventricular thalamus controls a central amygdala fear circuit. *Nature*.
- Rees, C.L., White, C.M., and Ascoli, G.A. (2017). Neurochemical Markers in the Mammalian Brain: Structure, Roles in Synaptic Communication, and Pharmacological Relevance. *Curr Med Chem* *24*, 3077-3103.
- Sanford, C.A., Soden, M.E., Baird, M.A., Miller, S.M., Schulkin, J., Palmiter, R.D., Clark, M., and Zweifel, L.S. (2017). A Central Amygdala CRF Circuit Facilitates Learning about Weak Threats. *Neuron* *93*, 164-178.
- Sun, Q., Li, X., Ren, M., Zhao, M., Zhong, Q., Ren, Y., Luo, P., Ni, H., Zhang, X., Zhang, C., *et al.* (2019). A whole-brain map of long-range inputs to GABAergic interneurons in the mouse medial prefrontal cortex. *Nature neuroscience*.
- Tremblay, R., Lee, S., and Rudy, B. (2016). GABAergic Interneurons in the Neocortex: From Cellular Properties to Circuits. *Neuron* *91*, 260-292.
- Wang, X., Zhang, C., Szabo, G., and Sun, Q.Q. (2013). Distribution of CaMKIIalpha expression in the brain in vivo, studied by CaMKIIalpha-GFP mice. *Brain Res* *1518*, 9-25.
- Wang, Y., Hu, P., Shan, Q., Huang, C., Huang, Z., Chen, P., Li, A., Gong, H., and Zhou, J.N. (2021). Single-cell morphological characterization of CRH neurons throughout the whole mouse brain. *BMC Biol* *19*, 47.
- Watabe-Uchida, M., Zhu, L., Ogawa, S.K., Vamanrao, A., and Uchida, N. (2012). Whole-brain mapping of direct inputs to midbrain dopamine neurons. *Neuron* *74*, 858-873.

REVIEWERS' COMMENTS:

Reviewer #1 (Remarks to the Author):

The authors have thoughtfully addressed my previous concerns. The revised manuscript is significantly improved, and I commended them for their efforts.

Reviewer #2 (Remarks to the Author):

The authors have addressed all of my concerns -- the manuscript is suitable for publication.

Reviewer #3 (Remarks to the Author):

The authors have fully addressed all previously raised concerns and suggestions.

Reviewer comments:

Reviewer #1:

Remark to author:

The authors have thoughtfully addressed my previous concerns. The revised manuscript is significantly improved, and I commended them for their efforts.

Reply:

We are glad to hear that we have thoughtfully addressed the previous concerns of the reviewer. Thank you for commenting that the revised manuscript is significantly improved and commending our efforts.

Reviewer #2:

Remark to author:

The authors have addressed all of my concerns -- the manuscript is suitable for publication.

Reply:

It's our pleasure to address all concerns of the reviewer, and thank you for your approval.

Reviewer #3:

Remark to author:

The authors have fully addressed all previously raised concerns and suggestions.

Reply:

We are pleased to address all the concerns and suggestion raised by the reviewer.